# Transformation of spatiotemporal dynamics in the macaque vestibular system from otolith afferents to cortex

Jean Laurens[1†], Sheng Liu[2†], Xiong-Jie Yu[1,3,4†], Raymond Chan[1], David Dickman[1], Gregory C DeAngelis[5], Dora E Angelaki[1]*

[1]Department of Neuroscience, Baylor College of Medicine, Houston, United States; [2]State Key Laboratory of Ophthalmology, Zhongshan Opthalmic Center, Sun Yat-sen University, Guangzhou, China; [3]Zhejiang University Interdisciplinary Institute of Neuroscience and Technology, Zhejiang University, Hangzhou, China; [4]Qiushi Academy for Advanced Studies, Zhejiang University, Hangzhou, China; [5]Deptartment of Brain and Cognitive Sciences, University of Rochester, Rochester, United States

**Abstract** Sensory signals undergo substantial recoding when neural activity is relayed from sensors through pre-thalamic and thalamic nuclei to cortex. To explore how temporal dynamics and directional tuning are sculpted in hierarchical vestibular circuits, we compared responses of macaque otolith afferents with neurons in the vestibular and cerebellar nuclei, as well as five cortical areas, to identical three-dimensional translational motion. We demonstrate a remarkable spatio-temporal transformation: otolith afferents carry spatially aligned cosine-tuned translational acceleration and jerk signals. In contrast, brainstem and cerebellar neurons exhibit non-linear, mixed selectivity for translational velocity, acceleration, jerk and position. Furthermore, these components often show dissimilar spatial tuning. Moderate further transformation of translation signals occurs in the cortex, such that similar spatio-temporal properties are found in multiple cortical areas. These results suggest that the first synapse represents a key processing element in vestibular pathways, robustly shaping how self-motion is represented in central vestibular circuits and cortical areas.

*For correspondence: angelaki@bcm.edu

[†]These authors contributed equally to this work

**Competing interests:** The authors declare that no competing interests exist.

## Introduction

Many sensory systems have been studied along their hierarchy, from primary receptor cells to cortical neurons. Such systematic analyses provide a foundation for comprehending the neural computations that convert early sensory activity into higher level constructs that underlie perception, action, and other cognitive functions. For the vestibular system, however, such analysis has been largely limited to reflex generation in the brainstem and cerebellum. On the other hand, there are multiple cortical areas that respond to vestibular stimuli, typically together with other sensory and motor signals (*Grüsser et al., 1990*; *Bremmer et al., 2002*; *Fukushima et al., 2006*; *Klam and Graf, 2006*; *Chen et al., 2010*, *2011a*, *2011b*, *2011c*; *Gu et al., 2006*, *2016*). How do the spatial and temporal properties of neurons in cortical areas differ from those in subcortical vestibular hubs?

Primary otolith afferents are spatially cosine-tuned and their temporal dynamics are broadly thought to encode translational acceleration (*Fernández and Goldberg, 1976a*, *1976b*, *1976c*; *Angelaki and Dickman, 2000*; *Jamali et al., 2009*). Neural response properties in the vestibular nuclei (VN), which receive the bulk of vestibular afferent projections outside the cerebellum (*Barmack, 2003*; *Newlands and Perachio, 2003*; *Angelaki and Cullen, 2008*), are different.

It has been proposed that spatio-temporal convergence of otolith afferents onto central VN cells results in complex, non-cosine-tuned properties, where spatial and temporal coding might not always be multiplicatively separable (*Angelaki, 1991*, *1992b*; *Angelaki et al., 1992*; *Angelaki, 1993*; *Angelaki et al., 1993*; *Bush et al., 1993*; *Angelaki and Dickman, 2000*; *Dickman and Angelaki, 2002*; *Chen-Huang and Peterson, 2006*, *2010*). Some studies have supported this prediction in the vestibular brainstem (*Angelaki et al., 1993*; *Bush et al., 1993*; *Angelaki and Dickman, 2000*; *Chen-Huang and Peterson, 2006*, *2010*), but direct comparisons with cortical responses (e.g., *Chen et al., 2011c*) have never been made.

Here, we use data obtained from transient translational displacements along multiple directions in three-dimensional (3D) space to compare the spatio-temporal response properties of neurons in the vestibular and rostral medial cerebellar nuclei (VN/CN; *Liu et al., 2013*) with responses in multiple cortical areas, including the parietoinsular vestibular cortex (PIVC; *Chen et al., 2010*), visual posterior sylvian area (VPS, a visual/vestibular convergent area just posterior to PIVC; *Chen et al., 2011b*), ventral intraparietal area (VIP; *Chen et al., 2011c*), dorsal medial superior temporal area (MSTd; *Gu et al., 2006*; *Takahashi et al., 2007*; *Gu et al., 2010*) and the frontal eye fields (FEF; *Gu et al., 2016*). In addition, we have also recorded from primary otolith afferent fibers from the vestibular subdivision of the eighth nerve in response to identical stimuli (*Yu et al., 2015*). We report a remarkable spatio-temporal transformation between otolith afferents and VN/CN cells, and this transformation determines the main response properties carried forward to cortical neurons.

## Results

### Data set, model composition and example model fits

The neural data set used to fit 3D spatio-temporal models consisted of the average temporal response profile (PSTH) of each neuron for 26 directions of translation corresponding to all possible combinations of azimuth and elevation angles in increments of 45° (*Gu et al., 2006*). The temporal waveform of the translational stimulus followed a Gaussian velocity profile (*Figure 1*, top), with corresponding biphasic acceleration (*Figure 1*, middle) and triphasic jerk (derivative of acceleration; *Figure 1*, bottom) components. We considered only cells with significant spatial and temporal response modulation, as detailed by *Chen et al. (2011a)* (see also Materials and methods). This inclusion criterion yielded a total of 27 otolith afferents (OA), 49 VN cells, 61 CN cells, 115 PIVC cells (from *Chen et al., 2010*), 66 VPS cells (*Chen et al., 2011b*), 139 MSTd cells (from *Gu et al., 2006*), 62 VIP cells (from *Chen et al., 2011c*) and 57 FEF cells (from *Gu et al., 2016*) (see *Table 1*).

Multiple models of varying complexity were fit to the PSTHs of each neuron (*Figure 1*). In its most general form, the standard model consisted of the sum of three response components, having temporal dynamics associated with velocity, acceleration and jerk. For each component, the temporal profile of the stimulus ($f_v(t - \tau_0)$, $f_a(t - \tau_0)$ or $f_j(t - \tau_0)$) was multiplied by a 3D spatial tuning function ($y_v(g_v(\theta, \phi))$, $y_a(g_a(\theta, \phi))$, $y_j(g_j(\theta, \phi))$; cosine tuning with an offset; see Materials and methods) and a weight ($W_v$, $W_a$ or $W_j$). This required four fitted parameters for each component. The sum of these three components was added to the resting discharge ($FR_0$), and a temporal delay term ($\tau_0$) was introduced (*Figure 1*). Thus, the maximum number of free parameters ('VAJ' model), was 14. For an easy comparison of the relative importance of the

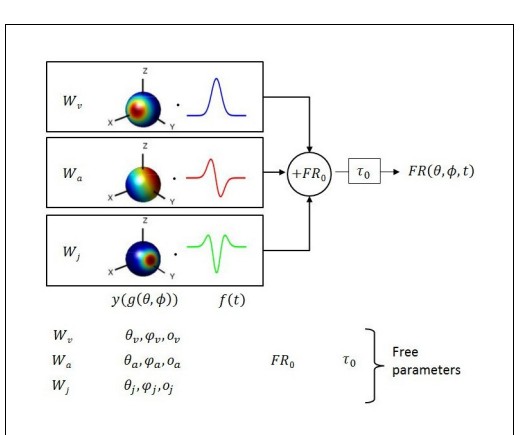

**Figure 1.** Schematic of model with velocity, acceleration and jerk components. The fitted function, $FR(\theta, \phi, t)$, is the sum of three components, each consisting of a weight $W$, a 3D spatial tuning function ($y(g(\theta, \phi))$ represented on a sphere) and a temporal response profile ($f(t)$), scaled and multiplied together. Spatial tuning functions illustrate that preferred directions need not be identical for each temporal component.

**Table 1.** Overview of the data. The table provides a list of the brain regions included in the current analyses, the number (m) of mon-keys used in each area (note that neurons were recorded in more than one area in some monkeys), the number (n) of neurons analyzed from each area and references to previous publications where technical details are provided. Neurons were either reported here for the first time (*new data*) or re-analyzed from previous publications (references in last column).

| Area name and abbreviation | | m | n | Methodological details and original publication |
|---|---|---|---|---|
| Otolith Afferent fibers, eighth cranial nerve | OA | 2 | 27 | New data/*Yu et al. (2015)* |
| Vestibular Nuclei | VN | 4 | 49 | New data/*Liu et al. (2013)* |
| Rostral medial Cerebellar Nuclei | CN | 5 | 61 | New data/*Liu et al. (2013)* |
| Parietoinsular Vestibular Cortex | PIVC | 2 | 115 | *Chen et al. (2010)* |
| Visual Posterior Sylvian area | VPS | 3 | 69 | *Chen et al. (2011a)* |
| Dorsal Medial Superior Temporal area | MSTd | 3 | 139 | *Gu et al. (2006, 2010); Takahashi et al. (2007)* |
| Ventral Intraparietal area | VIP | 3 | 62 | *Chen et al. (2011a)* |
| Frontal Eye Field | FEF | 3 | 57 | *Gu et al. (2016)* |
| Total | | 19 | 579 | |

three temporal components, we also present the normalized weights ($w_v$, $w_a$ and $w_j$), which are equal to the weights divided by $W_v + W_a + W_j$.

Simpler models, consisting of either single or double component contributions were also fit to the responses of each cell, as described above, and included Acceleration-only ('A'), Velocity-only ('V'), Jerk-only ('J'), Velocity+Acceleration ('VA'), Velocity+Jerk ('VJ') and Acceleration+Jerk ('AJ') models. Because different models have different numbers of free parameters (6 parameters for the single-component models and 10 for the double-component models), the relative quality of the different model fits was assessed using the Bayesian Information Criterion (BIC; *Schwarz, 1978*).

*Figure 2* illustrates example fits of the VAJ model for an OA (*Figure 2A*). The biphasic response PSTHs resembled the stimulus acceleration profile, with its amplitude being spatially modulated. Peak responses, with opposite signs, were observed during upward and downward motion. Furthermore, the cell responded weakly to stimuli with 0° elevation regardless of azimuth, corresponding to orthogonal directions relative to upward/downward motion. This response pattern is characteristic of cosine spatial tuning, where the response is modulated as a function of the cosine between the stimulus direction and the cell's preferred direction.

For the example OA, the acceleration component had a large weight (*Figure 2B*, red) and partial $R^2$ (*Figure 2C*, red); the jerk component had a lower, but still sizable, weight and partial $R^2$ (blue), whereas the velocity component was negligible (*Figure 2B–D*). Accordingly, the AJ model was selected as best describing this cell's response based on BIC analysis (even though *Figure 2* illustrates the fits and parameters of the VAJ model). Note that the spatial tuning of acceleration and jerk were very similar (color maps in *Figure 2E,F*), indicating that spatial and temporal properties are separable (i.e. one does not depend on the other). This property is reflected in the cell's separability index (Sep.I, *Figure 2C*, black; see Materials and methods) of 1. *Figure 2G* further summarizes the spatial tuning properties of all three components by plotting the their response offset against the amplitude of cosine tuning, each multiplied by the weight of the respective components. Since the acceleration and jerk responses are purely cosine-tuned, these components have zero offset while both the response offset and the cosine tuning of the velocity component are close to zero. The fitted temporal response components at three example stimulus directions are shown in *Figure 2G,H,I* (marked accordingly in the PSTHs of *Figure 2A*).

In contrast to otolith afferents, many central neurons exhibited distinct spatial tuning for the different temporal response components, as illustrated by data from an example VN neuron (*Figure 3A–I*). Here, all dynamic components contributed substantially to the cell's response (*Figure 3B,C*), and the VAJ model had the best BIC. Unlike otolith afferents, this VN cell had a strong velocity modulation, which was positive along the preferred direction (close to downward; *Figure 3D and H*, green). However, the velocity component also exhibited a small positive (rather than negative) response during horizontal and upward motion directions. Thus, the velocity response

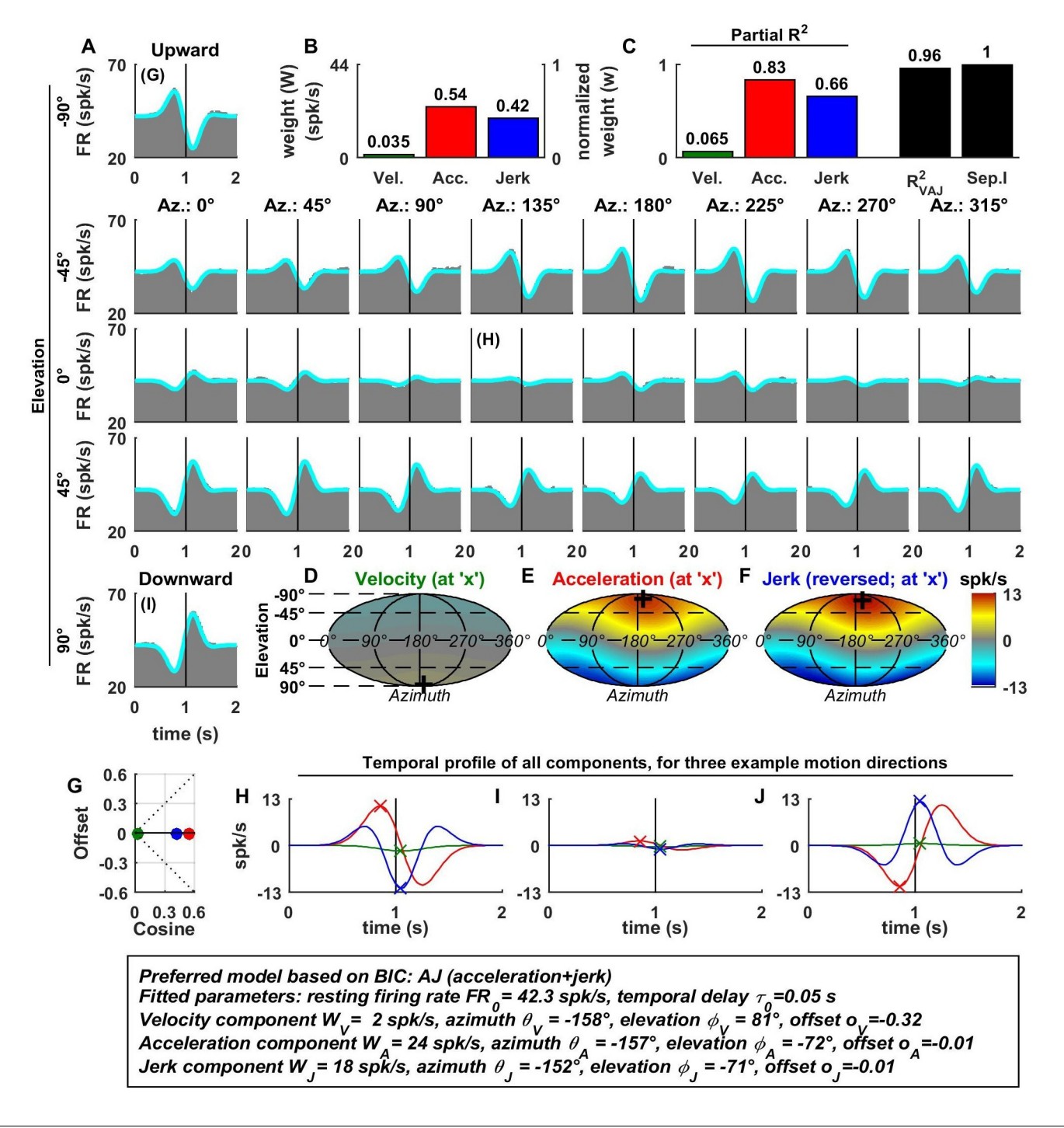

**Figure 2.** Spatio-temporal tuning of an example otolith afferent that was best fit by the AJ model. (A) PSTHs (gray) and model fits (cyan) for each of the 26 stimulus directions, defined by the corresponding (azimuth, elevation) angles. The vertical lines at t = 1s represent the timing of peak stimulus velocity. (B) Component weights for the fit of the VAJ model (left axis: raw weight in spikes/s, right axis: normalized weights, such that their sum = 1). (C) Partial $R^2$ of the three components, representing their contribution to the cell's firing; $R^2_{VAJ}$: goodness of fit of the full VAJ model; Sep.I: separability index, indicating how well the cell can be modeled using the same spatial tuning function for all three temporal response components (see Materials and methods). (D–F) Color intensity plots of the spatial tuning of velocity, acceleration and jerk components, respectively. For illustrative purposes, the sensitivities to velocity, acceleration and jerk, are multiplied by the peak amplitude of the respective temporal profiles such that the three spatial tuning functions are expressed in spikes/s. Therefore, (D) represents the contribution of the velocity component at the time of peak velocity (plus the delay $\tau_0$); (E) represents the contribution of acceleration at the time of peak acceleration (plus $\tau_0$); (F) represents the contribution of jerk at the time

*Figure 2 continued on next page*

*Figure 2 continued*

of peak jerk (plus $\tau_0$). The crosses in (D–F) indicate the preferred direction of each component. Note that a positive sensitivity to jerk (for upward jerk in this cell) corresponds to a negative response at t = 0 (see G) since jerk is negative at that time. (G) Offset ($w_0 * o_0$) versus cosine tuning amplitude ($w_0 * (1 - |o_0|)$) of the three temporal components. (H–J) Temporal profiles of the three components [velocity (green), acceleration (red) and jerk (blue) scaled by the respective component weight] for the stimulus directions indicated in A. The x marks illustrate the times for which the spatial tuning in D-F has been plotted.

component was positive in all directions (*Figure 3D*), which was modeled with a response offset greater than the amplitude of the cosine tuning (*Figure 3G*, green). The acceleration response was roughly cosine tuned, alternating from a positive response in downward movement directions to a negative response for upward movement directions (*Figure 3E*; see also *Figure 3H–J*, red); accordingly, its offset was close to zero (*Figure 3G*). Finally, the jerk response tuning was broad and mostly positive (*Figure 3F*; see also *Figure 3G–I*, blue).

Since the acceleration, velocity and jerk components had distinct spatial tuning, this VN neuron's separability index (0.73) was much lower than that for OAs, and the interplay of these components created strikingly distinct (i.e. mono-, bi- and tri-phasic) PSTH shapes for different stimulus directions (*Figure 3A*). For example, a combination of positive velocity and acceleration responses (*Figure 3G*) created a largely monophasic profile during downward motion (*Figure 3A*, bottom). In contrast, the PSTH was biphasic during upward motion, which was essentially driven by acceleration and jerk (*Figure 3I*). Note that the jerk temporal modulation was most obvious at horizontal stimulus directions, when the cosine-tuned acceleration was minimal (*Figure 3H*). The fit of the VAJ model captures a large portion of this diversity in the shapes of PSTHs across stimulus directions.

The preferred directions (PDs) for velocity and acceleration were 28° apart (difference in 3D, with peaks at [azimuth, elevation] = [333°, 49°], and [3°, 74°], respectively; *Figure 3D,E*, '+' in color contour plots). However, the jerk component had a nearly opposite preferred direction (at [azimuth, elevation] = [189°, −61°]; *Figure 3F*) and the difference with the preferred direction of the acceleration component was 167°. Note, however, that, given the strongly non-cosine-tuned properties of velocity and jerk (as captured by the non-zero offset parameters), PD was inadequate to fully characterize the spatial properties of most central cells.

Additional examples of cell responses and model fits (*Figure 3—figure supplements 1–6*) illustrate a diversity of non-linear response types encountered in central neurons. Remarkably, some neurons had either purely positive responses in all directions (*Figure 3—figure supplements 1* and *3*) or purely suppressive responses in all directions (*Figure 3—figure supplement 5*), whereas other cells responded only in a narrow range of motion directions (*Figure 3—figure supplements 4* and *6*). The VAJ model captured this variety of responses reasonably well, and therefore, an analysis of its parameters allowed us to draw quantitative conclusions from comparisons of cell tuning across neuronal populations.

## Summary of best-fitting models

Model fits were generally good (*Figure 4A*), with median $R^2$ values of 0.78 for OA (CI=[0.67–0.88]), 0.65 for VN (CI=[0.59–0.73]), 0.64 for CN (CI=[0.57–0.68]), 0.71 for PIVC (CI=[0.67–0.75]), 0.67 for VPS (CI=[0.62–0.71]), 0.59 for MSTd (CI=[0.55–0.64]), 0.66 for VIP (CI=[0.59–0.66]) and 0.59 for FEF (CI=[0.52–0.66]). The percentage of neurons that were best fit by each of the 7 (V, A, J, VA, VJ, AJ and VAJ) models is shown in *Figure 4B*. Remarkably, there was a sharp contrast between OAs and central cells. Forty-eight percent of OAs were best fit by the A model and 33% by the AJ model, whereas only 19% were best fit by the VAJ model. This indicates that acceleration responses are ubiquitous in OAs, whereas half of them respond to jerk and only a few to velocity. In contrast, 6% of central cells were best fit by the V model, 20% and 10% by the VA and VJ model, and 60% by the VAJ model, indicating that velocity responses were very common in central neurons. For comparison with previous work (*Chen et al., 2011c*), we also tested a more complex model in which the non-linear function assumed the form of an exponential. We found that this model performed only marginally better (*Figure 4—figure supplement 1*); thus, we did not consider it further in this study.

The pattern of results seen in *Figure 4B* was also reflected in the partial $R^2$ values (squared partial correlation coefficients, *Figure 4C–E*), which indicate how much each model component contributed

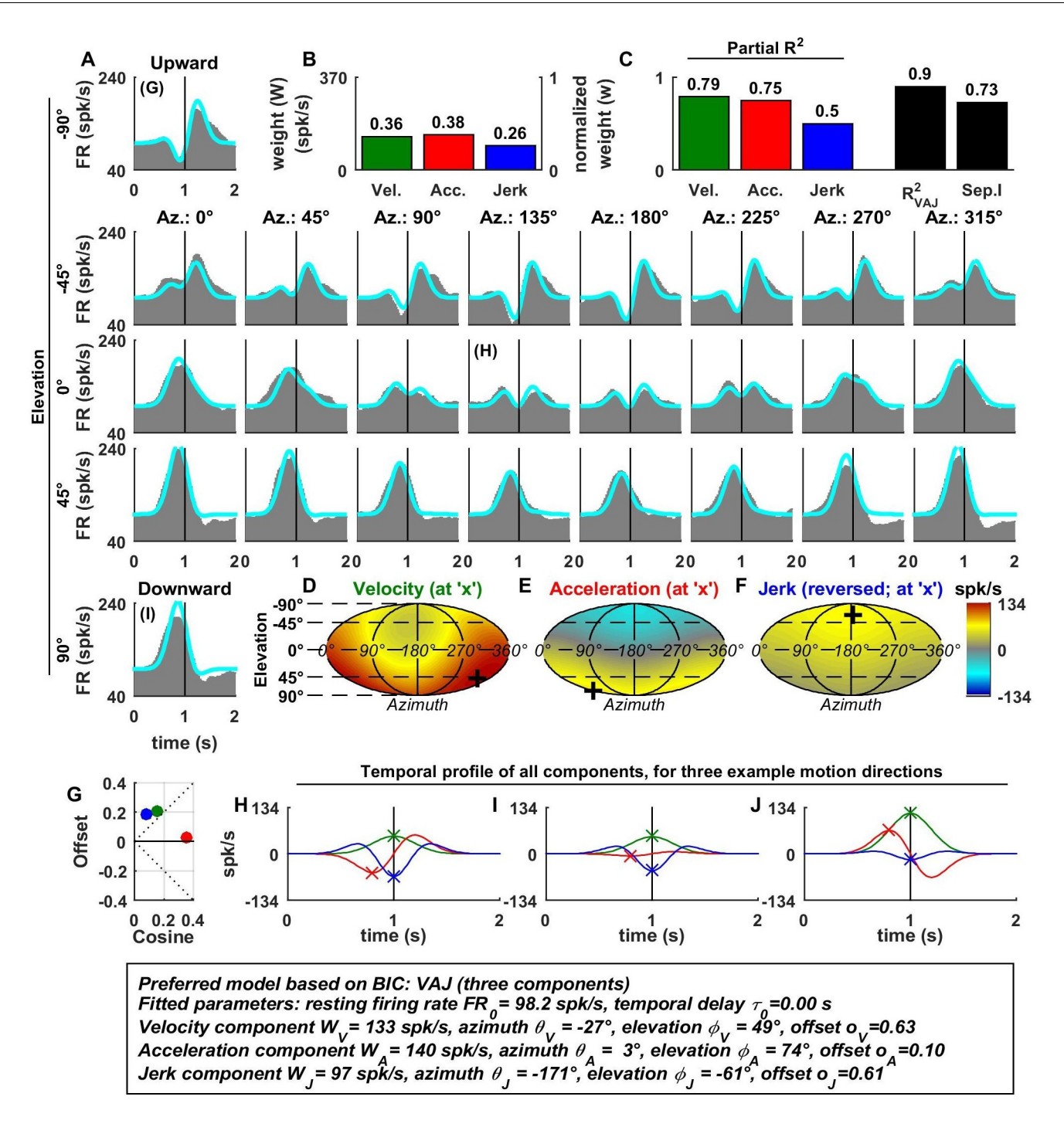

**Figure 3.** Spatio-temporal tuning and VAJ model fit for an example VN cell. Format as in *Figure 2*. Note that the neuron responds to translation in all directions (A), with different temporal response profiles that result from velocity, acceleration and jerk components having different spatial tuning (D–F), thus resulting in a relatively small separability index (C). Note also that the cell is characterized by nearly uniform tuning to velocity and jerk (D,F), which is modeled as a high offset combined with a relatively small cosine tuning amplitude (G, green and blue). Additional example cells are presented in *Figure 3—figure supplements 1–6*.

The following figure supplements are available for figure 3:

**Figure supplement 1.** Spatio-temporal tuning of an example CN cell.

*Figure 3 continued on next page*

*Figure 3 continued*

**Figure supplement 2.** Spatio-temporal tuning of an example PIVC cell, exhibiting a strong, cosine-like response to jerk.

**Figure supplement 3.** Spatio-temporal tuning of an example VPS cell, whose temporal response is dominated by velocity, exhibiting a nearly uniform response to motion in all directions (D; high offset and low cosine tuning amplitude in G, green) and a high separability index (=1), despite strongly non-linear responsiveness.

**Figure supplement 4.** Spatio-temporal tuning of an example MSTd cell, whose temporal response is dominated by velocity.

**Figure supplement 5.** Spatio-temporal tuning of an example VIP cell, exhibiting a decrease in firing in response to motion in all directions due to a uniform negative tuning to velocity and jerk.

**Figure supplement 6.** Spatio-temporal tuning of an example FEF cell, exhibiting a narrowly tuned response to upward directions, consisting of mostly velocity and acceleration components.

to neural responses. In OAs, acceleration had high partial $R^2$ values (median: 0.51, CI=[0.36–0.61]), while the jerk and velocity contributions were small (median partial $R^2$: 0.07, CI=[0.02–0.13] and 0.03, CI=[0.02–0.06], respectively). In central neurons, the velocity contribution was greatest (median partial $R^2$: 0.38, CI=[0.35–0.4]), followed by acceleration and jerk (median partial $R^2$ : 0.23, CI=[0.21–0.25] and 0.14, CI=[0.13–0.15], respectively).

Next, we will summarize in detail the model fits of otolith afferents, followed by a quantitative description of model parameters in other brain areas.

## Model fits to otolith afferent responses

As shown in *Figure 4B*, about half (n = 13) of OAs were best fit by the 'A' model (*Figure 5*, filled symbols), whereas the remainder (n = 14) were best fit by models of higher complexity ('AJ' and 'VAJ', open symbols in *Figure 5*). We found that these two groups had distinct firing properties. In particular 'A' afferents had a lower normalized coefficient of variation CV* (*Goldberg et al., 1984*) than 'AJ/VAJ' afferents (median 0.026, CI=[0.024–0.033] versus 0.06, CI=[0.04–0.08], Wilcoxon rank sum test, p=$10^{-3}$; *Figure 5A*). We also found that the higher the CV* the higher the weights for velocity, acceleration and jerk ($W_V - $ green, $W_A$-red, $W_J$-blue; *Figure 5B*, Spearman's rank correlation p<$10^{-2}$ for all three variables), as response gain increases with CV*. Importantly, the relative contributions of acceleration and jerk, as indicated by the normalized weights $w_A$ and $w_J$, (*Figure 5C*) were oppositely correlated with CV* (Spearman's rank correlation, p<$10^{-3}$ for both). Specifically, the normalized weight on acceleration declined with CV* while the normalized weight on jerk increased with CV*. In contrast, the contribution of velocity remained minimal (median = 0.06, CI=[0.05–0.08]) and did not correlate with CV* (Spearman's rank correlation, p=0.75; *Figure 5C*, green).

The preferred directions of the acceleration and jerk components were always aligned for OAs, as illustrated by the clustering of the A-J preferred direction differences (*Figure 5D*, black) close to 0° (median = 7°, CI=[4-9]). Thus, the two temporal components that dominated otolith afferent responses were spatially aligned, which is consistent with separable spatio-temporal tuning. Although velocity weights were overall small, the preferred direction of the velocity component relative to acceleration or jerk components (V-A or V-J) showed a significant dependence on CV* (both r = 0.7, p<0.01, Spearman's rank correlation; *Figure 5D*). For afferents with the most regular firing rates, the small velocity components tended to be orthogonal to acceleration and jerk components (V-A and V-J preferred direction differences ~90°). In contrast, the small velocity components of irregular otolith afferent responses tended to have direction preferences opposite to those of acceleration and jerk components (V-A and V-J relative angle differences ~180°; *Figure 5D*). The presence of this velocity component in OAs has not been identified before using sinusoidal stimuli (see Discussion).

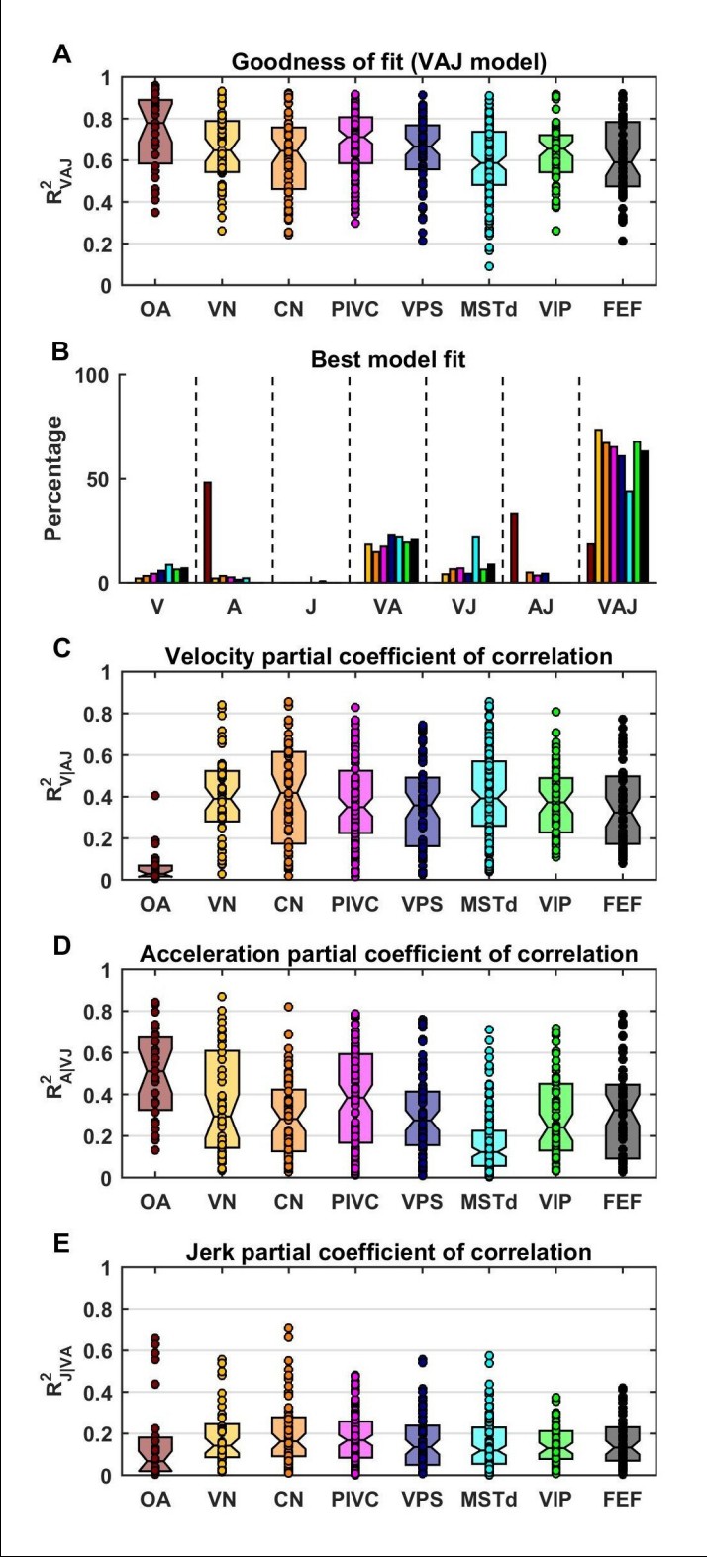

**Figure 4.** Summary of model fits. (**A**) Distribution of $R^2_{VAJ}$ values for each brain area. The boxes represent the median (center of the notch) and lower and upper quartiles of the population. Two medians are different at 5% level if the notches do not overlap. Individual data points are represented by circles. (**B**) Percentage of best model fits based on BIC. (**C–E**) Partial correlation coefficients of each of the three components, which reflect how much variance is accounted for by adding that component to the joint model of the other two components.

*Figure 4 continued on next page*

*Figure 4 continued*

Comparisons with model fits using an exponential spatial non-linearity function are shown in in *Figure 4—figure supplement 1*.

The following figure supplement is available for figure 4:

**Figure supplement 1.** Fitting performance using an exponential spatial non-linearity function.

## Model fits to central cell responses and comparison across areas

Identical models were fit to the responses of central neurons, with results summarized in *Figures 6–9*. We found large differences between the spatio-temporal properties of all central neurons, as compared to otolith afferents. The fitted velocity, acceleration and jerk component weights across areas are illustrated as *ternary plots*, which graphically depict the ratios of three variables that sum to a constant (in this case, the normalized weights, $w_v + w_a + w_j = 1$) as positions in an equilateral triangle (*Figure 6*). The value of each weight is 1 in one corner of the triangle and each weight decreases with increasing distance from this corner.

As described above, OAs (*Figure 6A*) respond predominantly to acceleration and display varying degree of sensitivity to jerk, while their response to velocity is minimal. Accordingly, they cluster between the A and J corners. The ternary plots of VN and CN neurons (yellow and orange, respectively) show a strikingly different pattern. Qualitatively, it is readily apparent that VN/CN cells carry substantially less acceleration signals, and much more velocity signals than otolith afferents. Quantitatively (see *Table 2*), the normalized acceleration weight (*Figure 6E*), which is high for otolith afferents, becomes significantly smaller for VN and CN neurons (p<10$^{-8}$, Wilcoxon rank sum test). In parallel, the velocity weights (*Figure 6D*) are significantly greater for VN and CN neurons, as compared to otolith afferents (p<10$^{-14}$, Wilcoxon rank sum test). There is little difference between jerk weights (*Figure 6F*) of VN/CN neurons and otolith afferents (p=0.03, Wilcoxon's rank sum test). Remarkably, despite dramatic differences in the relative weighting of acceleration and velocity signals between central cells and otolith afferents, VN and CN have similar properties overall, and there is no significant difference between weight values for the two areas (Wilcoxon's rank sum test, $w_v$: p=0.75; $w_a$; p=0.47; $w_j$: p=0.28).

As illustrated in *Figure 6B,C* and *Figure 6D–F*, the similarity in the relative weights among VN and CN neurons also extends to cortical areas (*Table 2*). The most noticeable difference among cortical areas is the slightly greater velocity weight in visual/vestibular multisensory areas (MSTd, VPS,

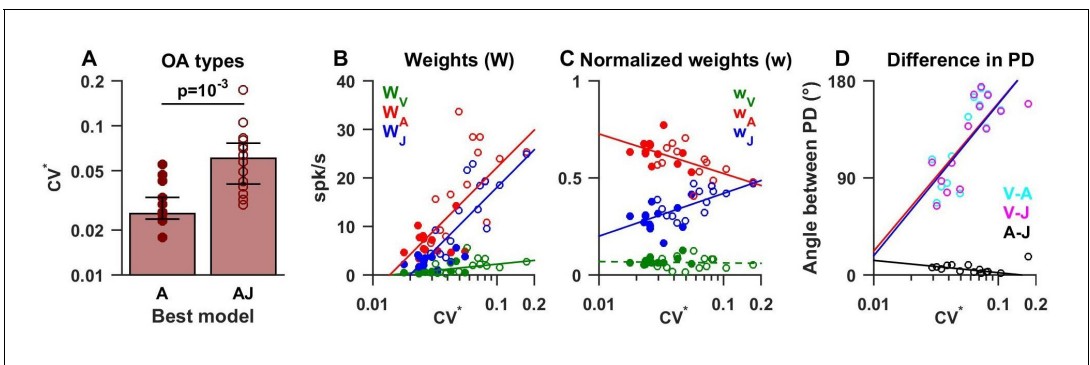

**Figure 5.** Summary of model fits for otolith afferents. (**A**) Difference in CV* between afferents that respond to acceleration only ('A') or acceleration, jerk and optionally velocity ('AJ' and 'VAJ'). (**B–C**) Raw and normalized weights for velocity, acceleration and jerk are plotted as a function of CV*. Filled symbols, 'A' afferents; open symbols: 'AJ' and 'VAJ' afferents. Note that weights for each OA were taken from the VAJ model such that all cells could be included in these plots. Solid lines: statistically significant (p<10$^{-2}$) type I regression lines. Broken lines: non-significant (p>0.05) regression lines. (**D**) The absolute difference between the 3D preferred directions of component pairs (A–J, V–A, V–J) plotted versus CV*. Here, angular differences are included only when both response components were significant for each cell. Symbols and lines as in **B,C**. Because A and J components were nearly aligned (black symbols), V-A (red) and V-J (blue) angular differences were similar for each cell.

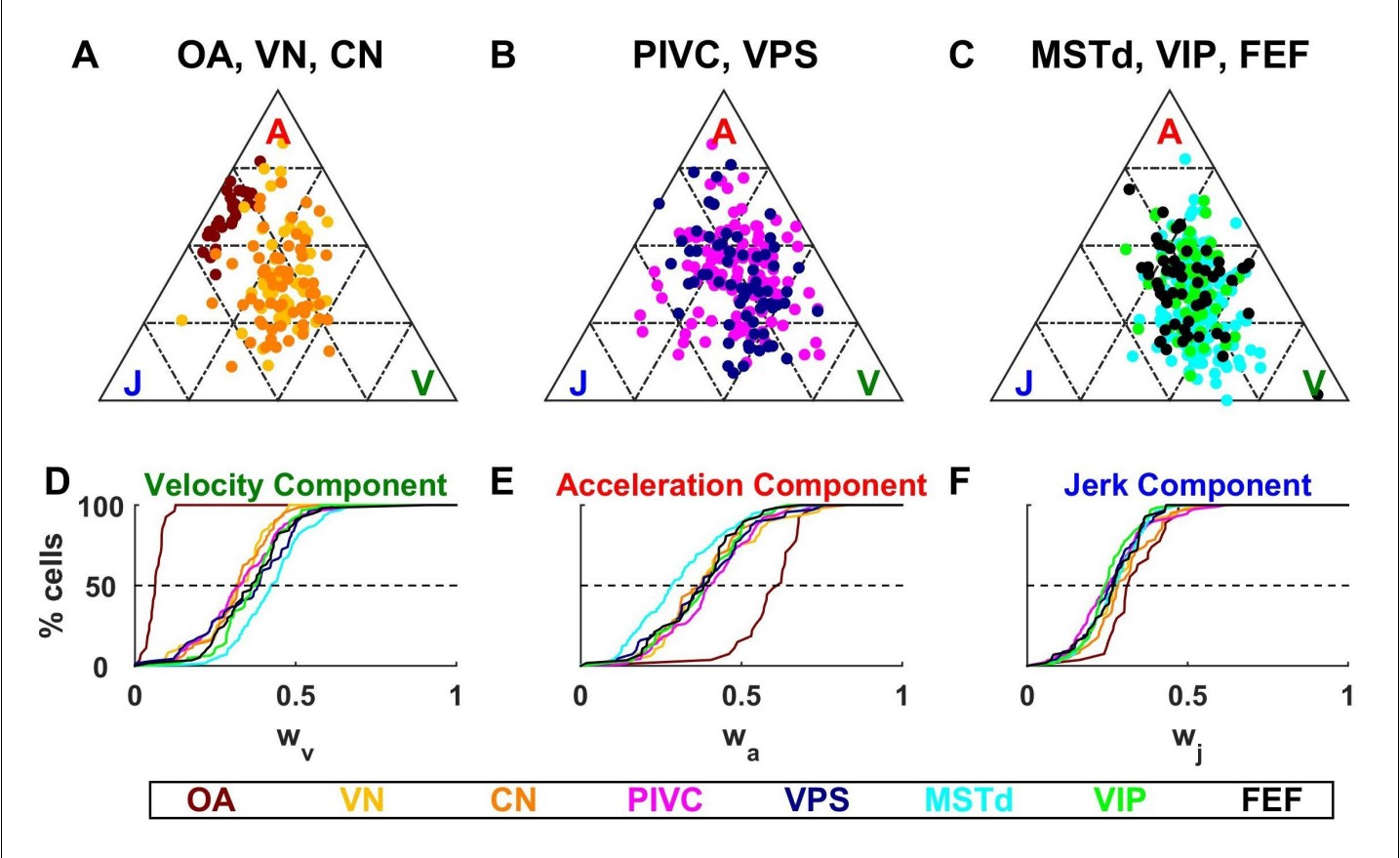

**Figure 6.** Dynamic components weights. (A–C) Ternary plots summarizing normalized weights for the acceleration, velocity and jerk components of VAJ model fits. Each data point represents the normalized velocity, acceleration and jerk weights for a single cell, color coded by brain area. (D–F) Cumulative distributions of normalized velocity (D), acceleration (E) and jerk (F) weights. Note the similarity in normalized velocity and acceleration weights across all central brain areas.

VIP and FEF; median: 0.39, CI=[0.38–0.41]), as compared to those without optic flow responsiveness (VN, CN, PIVC; median 0.32, CI=[0.31–0.35]) (*Figure 6D*, cold vs. warm colors, respectively; $p<10^{-9}$, Wilcoxon rank sum test). In fact, the normalized acceleration weight decreased, while the velocity weight increased from PIVC (median $w_a$= 0.41 [0.38–0.44] CI; median $w_v$= 0.33 [0.29 0.35] CI) to VIP (median $w_a$= 0.39 [0.35–0.42] CI; median $w_v$= 0.38 [0.34 0.4] CI) to MSTd (median $w_a$= 0.28 [0.26–0.32] CI; median $w_v$= 0.43 [0.4 0.45] CI), and these differences were significant (PIVC vs. MSTd: $w_v$: $p=3.10^{-9}$: $p = 3.10^{-8}$; VIP vs. MSTd: $w_v$: $p = 4.10^{-4} w_a$: $p = 3.10^{-3}$; PIVC vs. VIP: $w_v$: p=0.04; $w_a$: p = 0.3, rank sum test). Yet, these differences are substantially smaller than the transformation of vestibular translation signals between otolith afferents and VN/CN neurons.

To further characterize the transformation of spatio-temporal response properties from afferents to cortex, *Figure 7A* shows distributions of the separability index. OAs that were fit with the AJ or VAJ models (n = 14) generally have very high separability indices (median 0.98, CI=[0.97–0.99]), indicating that their strongest temporal components have similar preferred directions and spatial tuning. The remaining OAs carry only an acceleration component, a property that automatically confers a high separability index (median = 0.99, CI = [0.98–0.99]). In contrast, central brain areas have overall lower separability indices (median 0.89, CI=[0.88–0.9]; Wilcoxon rank sum test, $p<10^{-12}$). Some central cells are also spatio-temporally separable, but many others are not. For the latter neurons, each temporal component typically has a distinct directional tuning curve, resulting from a dependence of the temporal response profile on stimulus direction (e.g. *Figure 3*). Within central cells, we found a significant difference between the separability indices of VN/CN and cortical regions (median 0.82,

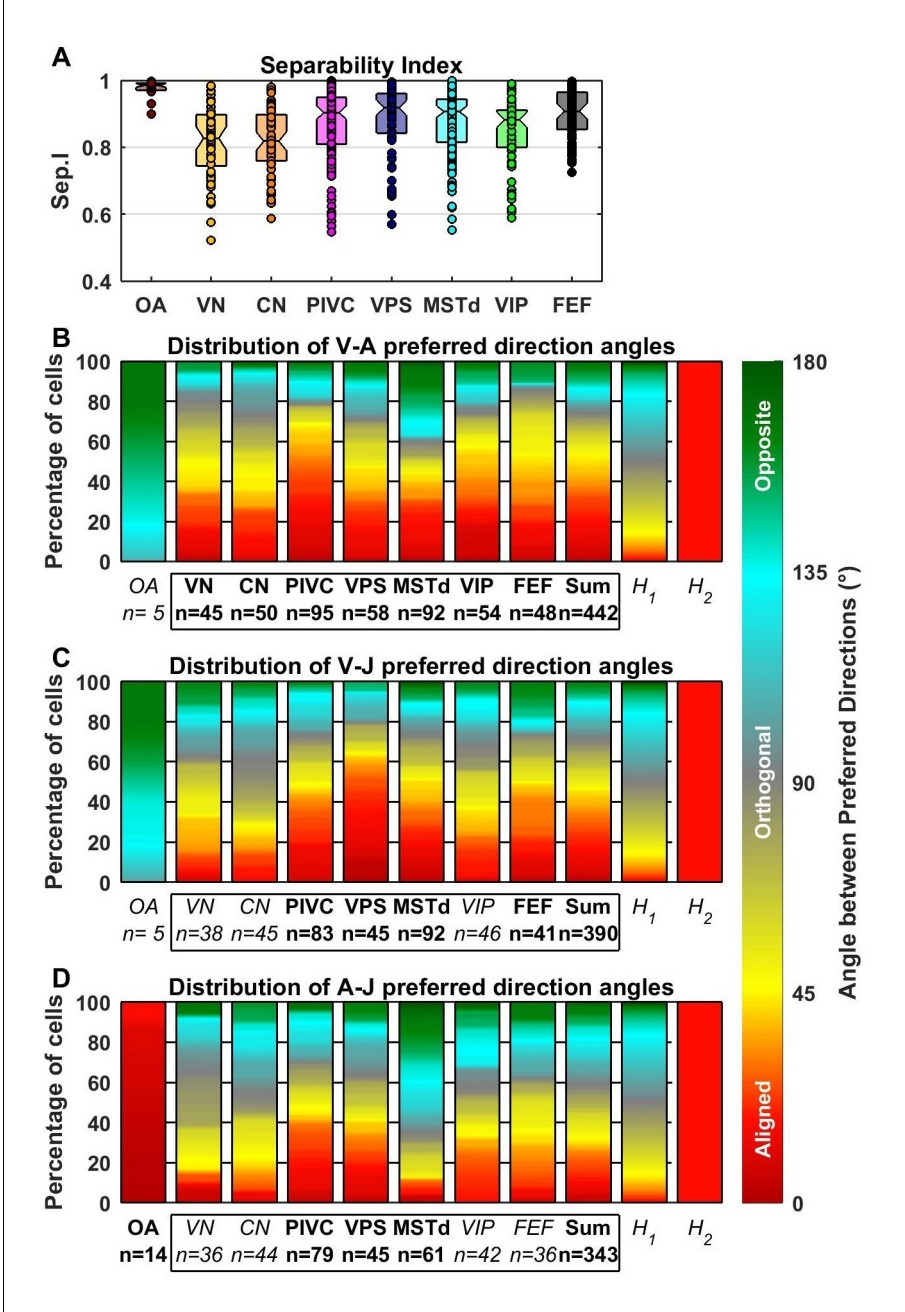

**Figure 7.** Comparison of preferred directions across dynamic components. (**A**) Distribution of the separability index across brain areas. (**B–D**) Summary of the angular difference between preferred directions of component pairs. The distribution of angles across the population is color-coded (red/yellow: aligned; green/cyan: opposite; grey: orthogonal). The 'sum' bar represents the distribution across all central brain areas (VN to FEF). For each comparison, only cells for which both components had significant spatial tuning have been considered (i.e. single-component best fits have not been included at all, whereas 2-component best fits have only been included in only one plot). The distribution $H_1$ represents the expected distribution if directions are distributed randomly on a sphere, in which case orthogonal pairs (e.g. from 85° to 95°) are more likely to occur than aligned or opposite pairs (e.g. from 0° to 10°). The distribution $H_2$ assumes only 'aligned' responses (like the A-J components of OAs). Bold/italic labels indicate distributions that are/are not significantly different from $H_1$ (top) (Kolomogorov-Smirnov Test; p-values indicated in **Table 3**). Distributions of the preferred direction of each dynamic component are shown in **Figure 7—figure supplement 1**.

The following figure supplement is available for figure 7:

*Figure 7 continued on next page*

*Figure 7 continued*

**Figure supplement 1.** Distributions of the PD of each dynamic component.

[0.81 0.84] CI versus 0.9, [0.89 0.91] CI, Wilcoxon rank sum test, $p<10^{-8}$). Differences in the separability index were much smaller among cortical areas (ANOVA, $F_{1,4}$ = 2.51, p=0.04).

The preferred directions of neurons were widely scattered in all brain regions, such that the distributions did not differ significantly from uniform in any individual region (*Figure 7—figure supplement 1*). *Figure 7B–D* shows the color-coded distribution of the angular difference in 3D preferred direction for each pair of significant temporal components. Bold/italic labels below each bar graph indicate distributions that are/are not significantly different from a uniform ('H₁') distribution. For comparison, a distribution that reflects purely aligned tuning ('H₂') is also shown.

Here too, the largest differences are seen between otolith afferents and central cells. The preferred directions for acceleration and jerk are aligned in all afferents that are tuned to these components (*Figure 7D*; see also *Figure 5D*, black). Because few afferents are tuned to velocity (based on BIC analysis), the angles between the velocity component and the jerk or acceleration components were not significantly different from uniform overall (even though they show a strong dependence on CV*; *Figure 5D*).

In contrast to otolith afferents, central neurons respond predominantly to the velocity and acceleration components of the stimulus (*Figure 4C,D*; *Figure 6*). We found that these components have aligned spatial tuning for a large subpopulation of central neurons (*Figure 7B*, red); thus, the distributions were all significantly different from uniform ('H₁'). In addition, they were all also clearly different from a purely aligned distribution (*Figure 7B–D*, right bars titled 'H₂'). This finding is consistent with the wide spread of separability indices (*Figure 7A*). Overall, similar observations also hold for the distributions of V-J and A-J preferred direction differences (*Figure 7C,D*). Note that the small separability index of VN/CN neurons (*Figure 4A*) is in line with a greater dispersion of differences in preferred directions between A, V and J components in VN and CN neurons (color-coded in red in *Figure 7B–D*). For example, only 19% of preferred directions (VA, VJ and AJ angles pooled) were aligned within 30° of each other in VN/CN versus 33% in other central brain regions.

Large differences between otolith afferent and central responses were also observed when comparing other model parameters (*Figure 8*; see also *Table 2*). Otolith afferents had the smallest modulation amplitude (peak-to-trough amplitude of the spatio-temporal response curves, computed from the VAJ model fit) as compared to all other brain areas (median = 10 spikes/s, CI=[8-18] for OA versus 43 spikes/s, CI=[41-46] for all central neurons, Wilcoxon rank sum test: $p=10^{-11}$;

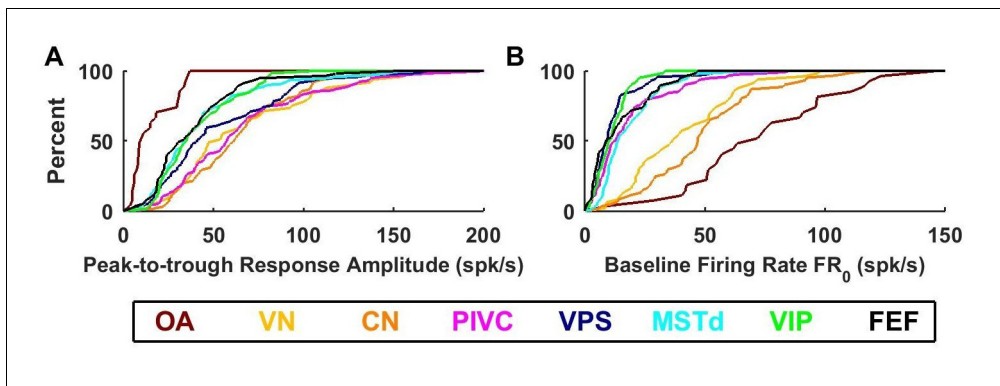

**Figure 8.** Summary of response amplitude and baseline firing rate parameters. Cumulative distributions of (**A**) peak-to-trough response amplitude (maximum across all directions), and (**B**) baseline firing rate (*FR₀*). OA: otolith afferents (brown); VN: vestibular nuclei (yellow); CN: cerebellar nuclei (orange); PIVC: parietoinsular vestibular cortex (red); VPS: visual posterior sylvian (blue); MSTd: dorsal medial superior temporal area (cyan); VIP: central intraparietal area (green). FEF: frontal eye fields (black).

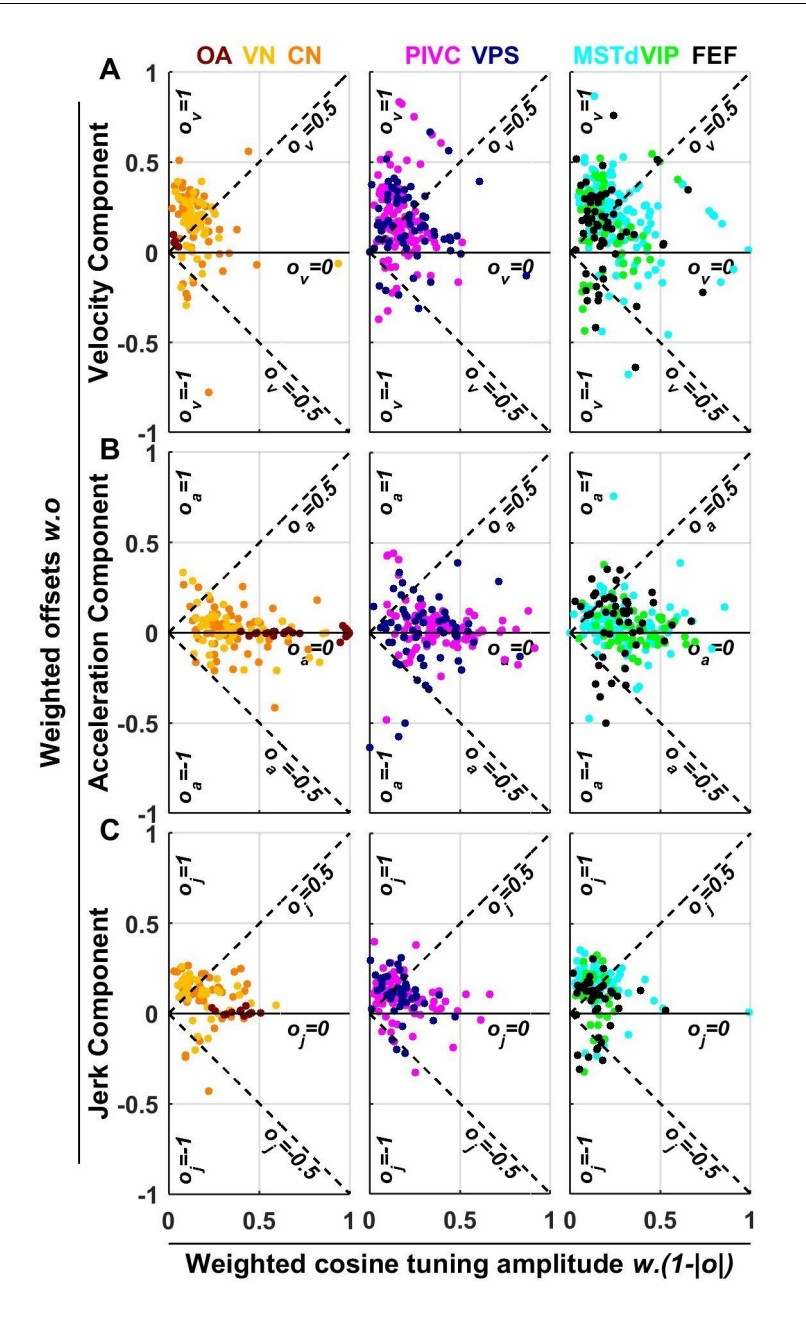

**Figure 9.** Summary of spatial tuning non-linearity for (A) velocity, (B) acceleration and (C) jerk. The spatial tuning curves $y_v(x)$, $y_a(x)$, $y_j(x)$ are modeled by adding an offset ($o_v$, $o_a$, $o_j$, respectively) to a cosine-tuned response $x$, i. e., $y_v(x) = o_v + (1 - |o_v|).x$ (the curves $y_a(x)$, $y_j(x)$ follow similar forms). The relative amplitudes of the response to velocity, acceleration and jerk depend on the weights $w_v$, $w_a$, $w_j$. Thus, the weighted cosine-tuned response components (abscissa) are: $w_v * (1 - |o_v|)$, $w_a * (1 - |o_a|)$, $w_j * (1 - |o_j|)$ and the weighted offset response components are: $w_v * o_v$, $w_a * o_a$, $w_j * o_j$ (ordinate). Purely cosine-tuned cells have zero offset ($o_v$, $o_a$, $o_j = 0$), while cells with positive or negative omnidirectional responses have a large offset and little modulation ($o_v$, $o_a$, $o_j \approx 1$ for positive responses, $o_v$, $o_a$ or $o_j \approx -1$ for negative responses). Distributions of the offset parameters are shown in *Figure 9—figure supplement 1*.

The following figure supplement is available for figure 9:

**Figure supplement 1.** Cumulative distributions of offset parameters for velocity (A), acceleration (B) and jerk (C) components across all brain areas.

**Table 2.** Model parameters based on area (median and CI).

| | | OA | VN | CN | PIVC | VPS | MSTd | VIP | FEF |
|---|---|---|---|---|---|---|---|---|---|
| $W_0$ (spikes/s) | | 23 [16-30] | 147 [122-172] | 144 [127-160] | 144 [129-159] | 122 [103-142] | 99 [88-110] | 99 [87-110] | 93 [78-108] |
| Peak-to-trough | | 16 [11-21] | 66 [55-78] | 65 [57-73] | 66 [58-73] | 54 [45-63] | 43 [38-48] | 41 [36-47] | 40 [32-47] |
| $FR_0$ (spikes/s) | | 74 [61-87] | 41 [34-47] | 49 [43-56] | 19 [15-22] | 12 [9-15] | 18 [16-21] | 11 [10-13] | 15 [11-18] |
| $w_v$ | | 0.07 [0.05–0.08] | 0.31 [0.28–0.34] | 0.32 [0.30–0.35] | 0.33 [0.31–0.36] | 0.35 [0.32–0.39] | 0.43 [0.41–0.44] | 0.37 [0.34–0.39] | 0.37 [0.33–0.40] |
| $w_a$ | | 0.60 [0.57–0.63] | 0.40 [0.36–0.45] | 0.37 [0.34–0.41] | 0.41 [0.38–0.43] | 0.38 [0.35–0.42] | 0.30 [0.28–0.33] | 0.38 [0.35–0.41] | 0.37 [0.33–0.40] |
| $w_j$ | | 0.34 [0.30–0.37] | 0.28 [0.25–0.31] | 0.30 [0.28–0.33] | 0.26 [0.24–0.28] | 0.26 [0.24–0.28] | 0.27 [0.26–0.28] | 0.25 [0.23–0.27] | 0.27 [0.24–0.29] |
| $o_v$ | | 0.58 [0.32–0.84] | 0.39 [0.26–0.51] | 0.44 [0.32–0.57] | 0.42 [0.34–0.50] | 0.38 [0.28–0.47] | 0.38 [0.32–0.44] | 0.33 [0.22–0.44] | 0.29 [0.16–0.42] |
| $o_a$ | | 0.00 [-0.01–0.01] | 0.06 [-0.02–0.14] | 0.06 [-0.01–0.14] | 0.07 [0.02–0.12] | 0.02 [-0.07–0.12] | 0.14 [0.09–0.20] | 0.06 [-0.01–0.13] | 0.17 [0.05–0.29] |
| $o_j$ | | 0.02 [-0.00–0.05] | 0.34 [0.22–0.46] | 0.35 [0.23–0.46] | 0.36 [0.29–0.44] | 0.38 [0.27–0.48] | 0.51 [0.45–0.58] | 0.43 [0.31–0.54] | 0.23 [0.07–0.39] |
| $\mu_0$ (s) | | 0.06 [0.05–0.07] | 0.04 [0.02–0.07] | 0.04 [0.02–0.07] | 0.04 [0.03–0.05] | 0.05 [0.02–0.07] | 0.14 [0.12–0.16] | 0.07 [0.05–0.10] | 0.06 [0.03–0.09] |
| Best fitting model | V | 0% | 2% | 3% | 4% | 6% | 9% | 6% | 7% |
| | A | 48% | 2% | 3% | 3% | 1% | 2% | 0% | 0% |
| | J | 0% | 0% | 0% | 0% | 0% | 1% | 0% | 0% |
| | VA | 0% | 18% | 15% | 17% | 23% | 22% | 19% | 21% |
| | VJ | 0% | 4% | 7% | 7% | 4% | 22% | 6% | 9% |
| | AJ | 33% | 0% | 5% | 3% | 4% | 0% | 0% | 0% |
| | VAJ | 19% | 74% | 67% | 66% | 62% | 44% | 69% | 63% |

**Table 3.** p-values of the Kolmogorov-Smirnov tests in **Figure 7**.

| | OA | VN | CN | PIVC | VPS | MSTd | VIP | FEF | Sum |
|---|---|---|---|---|---|---|---|---|---|
| V-A PD angles | 0.01 | $<10^{-4}$ | $<10^{-4}$ | $<10^{-4}$ | $<10^{-4}$ | $<10^{-4}$ | $<10^{-4}$ | $<10^{-4}$ | $<10^{-4}$ |
| V-J PD angles | 0.01 | 0.006 | 0.2 | $<10^{-4}$ | $<10^{-4}$ | $<10^{-4}$ | $<10^{-3}$ | $<10^{-4}$ | $<10^{-4}$ |
| A-J PD angles | $<10^{-4}$ | 0.14 | 0.1 | $<10^{-4}$ | $<10^{-3}$ | $<10^{-3}$ | 0.02 | 0.002 | $<10^{-4}$ |

**Figure 8A**). Furthermore, modulation amplitude varied significantly across central brain regions (one-way ANOVA, $F_{6,545} = 10$, p=$1.5*10^{-10}$). VN, CN and PIVC neurons were the most responsive (**Figure 8A**, warm colors), with aggregate modulation amplitude (median = 58 spikes/s, CI=[54-63]) greater than visual/vestibular multisensory areas VPS, MSTd, VIP and FEF (**Figure 8A**, cold colors; median: 35 spikes/s, CI=[33-39], Wilcoxon's rank sum test: p<$10^{-13}$). The reverse pattern of results was observed for the baseline response (one-way ANOVA across all brain regions: $F_{7,571} = 74$, p<$10^{-75}$). $FR_o$, which was greatest in otolith afferents (median = 72 spikes/s, CI=[54-90], **Figure 8B**, brown), intermediate in VN/CN (median = 45 spikes/s, CI=[38-49], **Figure 8B**, yellow/orange), and smallest in cortical areas (median = 12 spikes/s, CI=[11-13], **Figure 8B**). Thus, small gain modulation and high baseline firing rates in primary otolith afferents are converted into high gain modulation and low baseline firing rates in central brainstem, cerebellar and cortical neurons. Note, however, that this baseline firing rate could include a steady-state response to gravity (since animals were always oriented upright), which is unequivocal in otolith afferents but reduced across the population in brainstem, cerebellar and cortical neurons (**Angelaki et al., 2004**; **Liu and Angelaki, 2009**; **Liu et al., 2011**; **Shaikh et al., 2005**). An interesting picture also emerges when evaluating the spatial tuning, as summarized next.

## Spatial tuning curves

As illustrated in **Figures 2** and **3** and **Figure 3—figure supplements 1–6**, neuronal responses often showed non-linear spatial tuning that could be modeled by adding an offset to a cosine tuning function. The relative importance of offset and cosine-tuned responses are summarized in **Figure 9**. As illustrated in **Figure 2**, OAs mainly encode acceleration and jerk with cosine tuning. Accordingly, the offset parameters of the acceleration and jerk components cluster around zero for OAs (**Figure 9B, C**). In contrast, the weak velocity components of OAs display higher offsets (median offset parameter $o_v$ = 0.65, CI = [0.35–0.8]; **Figure 9A**).

Acceleration responses in central brain regions displayed more variable offsets, which nonetheless clustered around zero (median offset parameter $o_a$ = 0.08, CI = [0.06–0.11], **Figure 9B**). In absolute value, the cosine tuning amplitude was greater than the offset magnitude (i.e. $|o_a|<0.5$) for 88% of central neurons. These results indicate that acceleration responses tends to remain cosine-tuned throughout the brain.

In contrast, velocity responses usually displayed large positive offsets in all central brain regions (**Figure 9A**): the median offset parameter $o_v$ increased to 0.5 (CI = [0.47–0.52], rank sum test compared to $o_a$: p=$10^{-43}$; **Figure 9—figure supplement 1A**) and 50% of cells exhibited positive offsets greater than the amplitude of their cosine tuning component (i.e. $o_v>0.5$). Interestingly, a sizeable fraction (6%) of central cells exhibited negative offsets greater than their tuning (i.e. $o_v< -0.5$), leading to omnidirectional inhibitory responses (as in **Figure 3—figure supplement 5**). Finally, we found that jerk responses in central neurons also exhibited positive offsets (median 0.49, CI=[0.46 0.52]; **Figure 9C**) comparable to velocity responses.

## Position temporal modulation

The VAJ model has been based on the main response components encountered in all brain areas. We also fitted neuronal responses with a PVAJ model that included a position component, in addition to velocity, acceleration and jerk. For each cell, we extracted the amplitude of the position response component (**Figure 10A**, inset, black) and the total peak to peak modulation (**Figure 10A**, inset, red) for each spatial direction. The maximum total modulation (across all spatial directions) is plotted versus the maximum position modulation in **Figure 10A**. We define the position ratio as the

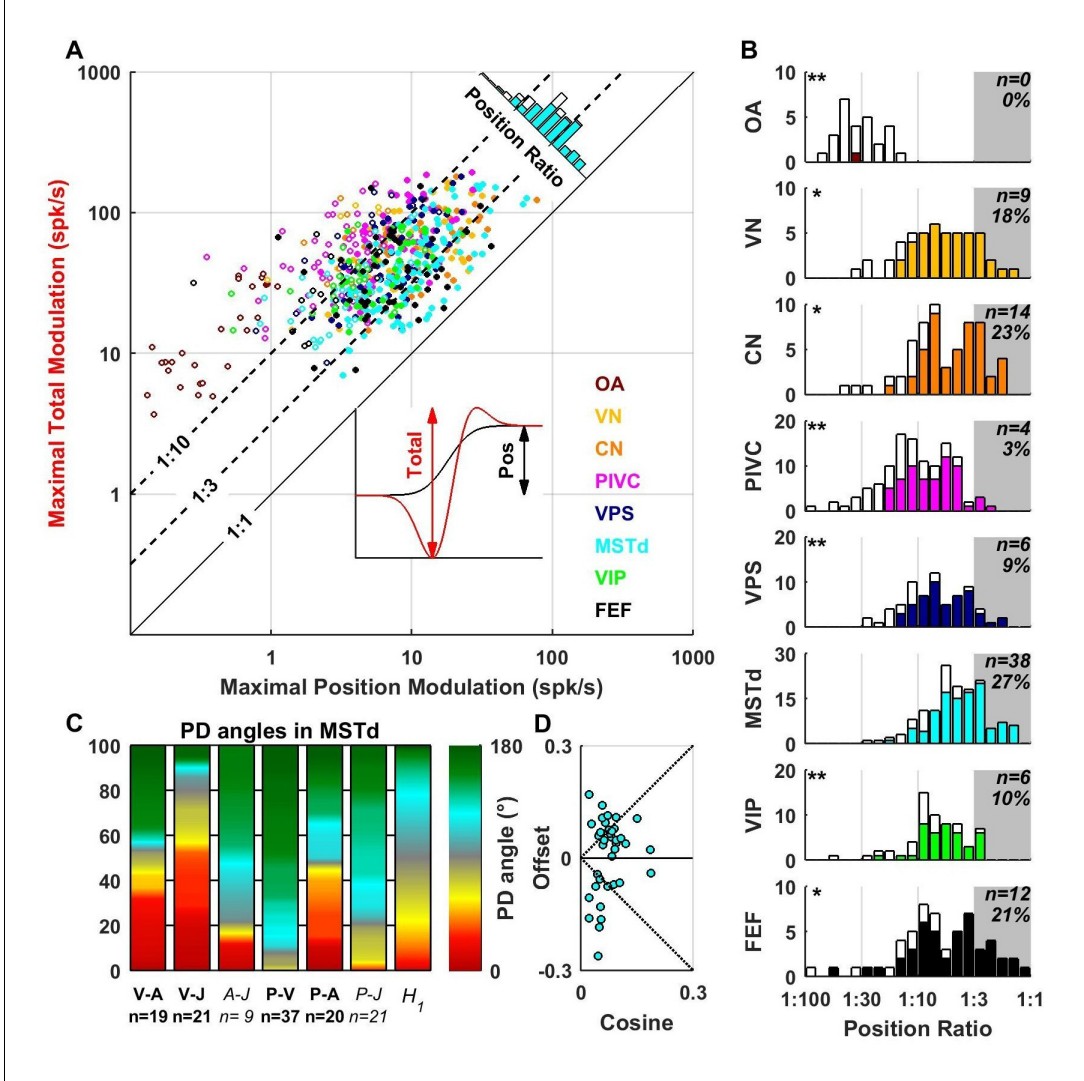

**Figure 10.** Contribution of position signals to vestibular responses. (A) Scatter plot of the maximum total peak to peak modulation vs. the maximal position modulation (see inset; computed across all spatial directions for each cell). We define the position ratio as the ratio of these two modulations. The diagonal and the two broken lines represent position ratios of 1:1, 1:3 and 1:10, respectively. (B) Position ratio in all brain areas (the histogram in MSTd is also represented as the marginal cyan distribution in A). The number and percentage of cells with a position ratio higher than 1:3 (grey region) is indicated for each area. All cells that are not significantly tuned to position are represented by open symbols and bars in A and B (and excluded from C,D). (C) Analysis of the angles between the PD of the position (P), velocity (V), acceleration (A) and jerk (J) components (as in *Figure 7*) for area MSTd. Only cells for which the position ratio is between 1:3 and 1:1 are included (other brain areas are not considered due to the lower numbers of cells). The P and V components have opposite PDs (green) for most MSTd neurons. In contrast, the distribution of P-A angles is symmetric, and similar numbers of cells have aligned and opposite PDs. Accordingly, the distribution of V-A angles is also symmetric. (D) Cosine tuning and offset components of position responses in MSTd (cells with position ratio greater than 1:3). The cells form two groups, one with positive offsets (26/39 cells, median offset = 0.46, [0.41 0.55] CI) corresponding to omnidirectional excitatory responses and another with large negative offsets (13/39 cells, median offset = −0.65, [-0.79–0.49] CI) corresponding to omnidirectional inhibitory responses.

ratio of these two modulations. The diagonal and the two broken lines represent position ratios of 1:1, 1:3 and 1:10, respectively. Cells with a strong position response have a position ratio close to 1:1 and appear close to the diagonal, whereas cells with a weak position response appear far above the diagonal.

Many central neurons had significant position contributions (*Figure 10A,B*, filled symbols and bars): OA: 1/27 (4%), VN: 41/49 (84%), CN: 47/61 (77%), PIVC: 63/115 (55%), VPS: 51/69 (74%), MSTd: 108/139 (78%), VIP: 40/62 (65%), FEF: 45/57 (79%). Furthermore, the percentage of neurons

with normalized position weights greater than 1:3 varied across areas (*Figure 10B*, gray-highlighted areas). In agreement with the findings of *Chen et al. (2011c)*, the position contribution was larger in MSTd, as compared to PIVC and VIP (areas for which the distribution is significantly different from MSTd are indicated by stars; *p<0.01; **p<0.001; two-sample Kolmogorov-Smirnoff tests). In addition to MSTd (cyan), we also found prominent position responses in VN (yellow), CN (orange) and FEF (black). The spatial properties of the position component were generally similar to those of the velocity component (shown for MSTd in *Figure 10C,D*), showing both inseparability and non-linear spatial tuning. Note, however, that future studies should record pre- and post-stimulus firing rates for a longer duration, such that these position responses can be quantified more reliably.

## Discussion

To understand the spatio-temporal processing of translational motion along the hierarchy of the vestibular system, from primary sensory neurons to subcortical and central cortical structures, we have performed a systematic comparison of spatio-temporal response properties to identical 3D naturalistic stimuli. Our analysis has revealed remarkably different spatio-temporal properties between central neurons and primary otolith afferents. Whereas otolith afferents encode mostly linear acceleration, central neuron responses are dominated by components related to stimulus velocity. Furthermore, central neurons also encode jerk (the derivative of linear acceleration), as well as position (displacement). However, this position response component needs to be further investigated in future studies using stimuli that allow a precise quantification of the pre-stimulus and post-stimulus firing rates (see Materials and methods).

The observed increase in velocity contribution to the spatio-temporal properties of central cells, as compared to otolith afferents, was accompanied by the presence of non-linearities. Whereas the linear acceleration component of central cells continued to be, like otolith afferents, mostly cosine-tuned, the velocity component included substantial omni-directional contributions, and jerk had intermediate properties. Thus, central neurons, like afferents, continued to carry information about the direction of linear acceleration, but they also encoded another type of signal: the presence or absence of translational motion without regard for direction. This component, which was strong in both the cortex and vestibular/cerebellar nuclei, cannot be characterized with the steady-state sinusoidal stimuli that have traditionally been the stimuli of choice in the vestibular system.

In contrast to the large differences in the types of signals encoded by otolith afferents versus their target cells in the vestibular nuclei, we found only moderate differences in the relative contributions of velocity, acceleration and jerk among central subcortical (vestibular and cerebellar nuclei) and cortical neurons. However, there were systematic differences in both the baseline firing and amplitude of stimulus-driven responses. Otolith afferents and vestibular/cerebellar nuclei neurons had high baseline firing compared to cortical neurons. Furthermore, response amplitude to an identical stimulus was lowest in otolith afferents and greatest in vestibular/cerebellar nuclei, PIVC and VPS – the areas known to form the core of the central vestibular system (*Angelaki et al., 2009*; *Lopez and Blanke, 2011*). In contrast, response amplitude is lower in MSTd, VIP and FEF, which are visual/vestibular multisensory areas. This result is consistent with *Chen et al. (2011c)*, who speculated that the vestibular pathway might go from OA to VN/CN to PIVC to VIP to MSTd. However, although modulation strength remains high in VN/CN and PIVC, it is nevertheless reduced in multisensory visual areas MSTd, VIP and FEF. Note that, although otolith afferents have low peak-to-trough modulation, their information content is high because of the high firing rate regularity (*Jamali et al., 2013*).

It is important to point out that *Chen et al. (2011c)* compared neural responses in a more limited number of cortical areas during motion within a single plane and reported a gradual shift from acceleration dominance in PIVC to velocity dominance in MSTd, as well as stronger position contributions in MSTd than PIVC or VIP. The present findings are consistent with both conclusions. However, when the full 3D properties are considered in the present analyses, we find that the spatio-temporal tuning properties of individual cortical areas also exhibit remarkable similarities. Note also that the model of *Chen et al. (2011c)* did not include a jerk component, which may also contribute to the results. Purkinje cells in the cerebellar vermal lobules X and IX, which were also previously fitted using the planar model (*Yakusheva et al., 2013*), also showed similar properties to central brain areas when analyzed using the full 3D model used here (unpublished). In summary, these results suggest that the signal transformation that takes place between sensory afferents and their targets in the

brainstem and cerebellum represents a key processing element in the vestibular system. Using identical passive stimuli, cortical areas show moderate differences, as compared to the response properties in the brainstem and cerebellum.

Understanding the central processing of translation signals is complicated for at least two reasons. First, afferents from a single otolith organ differ in both spatial and temporal response properties (*Fernández and Goldberg, 1976a*, *1976b*, *1976c*; *Angelaki and Dickman, 2000*; *Yu et al., 2012*; *Jamali et al., 2013*) and extensive convergence in the vestibular nuclei (*Kushiro et al., 2000*; *Straka et al., 2002*; *Uchino et al., 2001*; *Uchino and Kushiro, 2011*) creates complex spatio-temporal convergence properties (*Angelaki et al., 1992*; *Bush et al., 1993*; *Angelaki and Dickman, 2000*; *Dickman and Angelaki, 2002*; *Chen-Huang and Peterson, 2006*, *2010*).

Second, otolith afferents cannot distinguish tilt relative to gravity from translational acceleration. However, this *tilt/translation ambiguity* has been resolved in the cortex, where neurons that are selective for inertial (translational) accelerations are found (PIVC: *Liu et al., 2011*; MSTd: *Liu and Angelaki, 2009*). This property of cortical neurons is likely inherited from subcortical signals, as there is growing evidence that this computation is implemented through otolith/canal convergence in the brainstem and cerebellum (VN/CN: *Angelaki et al., 2004*; *Liu et al., 2013*; caudal cerebellar vermis: *Yakusheva et al., 2007*; *Laurens et al., 2013a*, *2013b*). It is thus possible that the large differences in spatio-temporal response properties that we have found between otolith afferents and VN/CN neurons arise from neural computations that resolve the tilt/translation ambiguity, a vital and critically important function for spatial orientation.

## Dynamic properties of otolith afferents

Otolith afferents have been previously characterized in terms of preferred direction in three-dimensions (*Fernández and Goldberg, 1976a*; *Tomko et al., 1981*; *Yu et al., 2012*) and response dynamics (*Anderson et al., 1978*; *Fernández and Goldberg, 1976b*; *Goldberg et al., 1990*; *Si et al., 1997*; *Angelaki and Dickman, 2000*; *Purcell et al., 2003*; *Jamali et al., 2009*). These two properties are separable, with the activity of each afferent fiber being determined by the product of a temporal 'transfer' function and a cosine-tuned spatial function (*Fernández and Goldberg, 1976a*, *1976b*, *1976c*). As a result of this separability, otolith afferents have the same response dynamics along different spatial directions.

It has long been recognized that otolith afferents differ in their response dynamics based upon the regularity of their spontaneous discharge (*Anderson et al., 1978*; *Angelaki and Dickman, 2000*; *Blanks and Precht, 1978*; *Fernández and Goldberg, 1976b*, *1976c*; *Goldberg et al., 1990*; *Jamali et al., 2009*, *2013*; *Purcell et al., 2003*). Indeed, we found that normalized jerk weights for otolith afferents increased with CV*, whereas acceleration weights decreased with CV*. Thus, the most regular afferents encoded pure acceleration, whereas more irregular afferents encoded mixtures of acceleration and jerk. These findings are consistent with previous studies reporting phase leads and gain advances in less regularly-firing afferents during sinusoidal stimulation (*Anderson et al., 1978*; *Angelaki and Dickman, 2000*; *Fernández and Goldberg, 1976b*; *Goldberg et al., 1990*; *Jamali et al., 2013*; *Purcell et al., 2003*; *Si et al., 1997*). Less regular-firing afferents also included a small but consistent velocity component, whose preferred direction relative to that of the jerk and acceleration components is correlated with CV*. Such a response component has not been described previously using sinusoidal stimulation.

## Spatio-temporal convergence – theoretical predictions and previous experimental findings using sinusoidal stimuli

Because VN neurons receive extensive convergence from otolith afferents that vary in their 3D spatial and temporal properties, a computational challenge arises. How would the properties of central neurons transform this information into signals that can be used by the rest of the brain? Theoretical analysis of such convergence showed that central neurons receiving otolith afferent convergence should, in general, exhibit spatio-temporal convergence properties, where spatial and temporal coding might not necessarily be separable; this results in complex spatial (non-cosine-tuned) properties, where central neurons carry different proportions of velocity, acceleration and jerk signals along different spatial directions (*Angelaki, 1992a*, *1992b*, *1992c*; *Schor and Angelaki, 1992*; *Angelaki et al., 1993*; *Angelaki and Dickman, 2000*). Evidence for simultaneous coding of both

translational acceleration and jerk was indeed demonstrated in VN responses of decerebrate rodents (*Angelaki et al., 1993*).

It took several more years until some experimental evidence for spatio-temporal convergence properties was provided for neurons in the primate VN (*Angelaki and Dickman, 2000*; *Chen-Huang and Peterson, 2006*, *2010*) and CN (*Shaikh et al., 2005*). However, to our knowledge, all previously published studies of responses of VN/CN neurons to translation used sinusoidal stimuli. There are many reasons why such stimuli are problematic. First, VN and CN responses are characterized by non-minimum phase properties, i.e., phase and gain changes do not parallel each other, thus complicating identification of temporal dynamics using sinusoids (*Angelaki and Dickman, 2000*; *Dickman and Angelaki, 2002*; *Shaikh et al., 2005*). Second, responses proportional to the temporal derivative and integral of translational acceleration cannot be easily separated using sinusoidal stimuli (e.g., a phase difference of 90° relative to acceleration could reflect either velocity coding in one direction or jerk coding in the opposite direction). The traditional approach of testing sinusoidal responses at different frequencies to distinguish velocity from jerk contributions using linear systems analyses is complicated by the presence of non-minimum phase properties, thus making system identification challenging (*Angelaki and Dickman, 2000*; *Dickman and Angelaki, 2002*; *Shaikh et al., 2005*). Furthermore, if indeed response dynamics vary with stimulus direction, a complete characterization of the spatio-temporal properties requires testing cells at multiple frequencies and multiple directions in 3D (*Chen-Huang and Peterson, 2006*, *2010*). Third, non-linearities in spatio-temporal tuning, as reported here (see also *Chen et al., 2011c*; *Massot et al., 2012*) can violate the assumptions of traditional linear systems analysis, thus rendering previous conclusions based on steady-state frequency analysis problematic.

These limitations have been circumvented here, for the first time, by the use of 3D transient stimuli and a comparison of neural responses in multiple subcortical and cortical areas. Importantly, in contrast to *Chen et al. (2011c)*, who fit cortical responses during translation in a single plane, we have developed models that characterize cell responses in 3D. With the exception of *Chen-Huang and Peterson (2010)*, who characterized 3D responses in the VN using sinusoidal stimuli (see limitations above), nothing is known about spatio-temporal convergence in any brain region using transient stimuli. Yet, as previously also emphasized by *Chen-Huang and Peterson (2010)*, understanding the full extent of the underlying spatio-temporal computations is only achievable when data are analyzed in 3D, and conclusions drawn by characterizing responses along a single direction or within a single plane can be rather incomplete. Our current findings have provided the first such quantification of 3D spatio-temporal convergence properties using transient stimuli along many spatial directions.

In contrast to cosine-tuning of otolith afferents, central neuron responses often show strongly nonlinear and rectified responses to both translational velocity and jerk. Most central cells show omni-directional tuning components. The existence of broad or uniform tuning, as opposed to cosine tuning, suggests that many afferents converge onto one central neuron. Indeed, broadly tuned neurons respond to motion in directions that are orthogonal to their PD. In contrast, in OA, all directions that are orthogonal to the PD are null directions.

These results suggest that the transformation of responses from sensory afferents to brainstem neurons represents a key processing element, possibly because it is coupled to the resolution of a tilt/translation ambiguity that represents a critically important function for spatial orientation. Although there is no direct evidence yet for such a link, theoretical solutions to the ambiguity problem require strong non-linearities (*Borah et al., 1988*; *Merfeld, 1995*; *Glasauer and Merfeld, 1997*; *Merfeld et al., 1999*; *Angelaki et al., 1999*, *2004*; *Laurens and Droulez, 2007*; *Laurens and Angelaki, 2011*).

## Functional implications

The functional significance of the observed spatio-temporal transformation that exists between otolith afferents and VN/CN cells, as well as all other cortical regions examined, remains to be determined. Still, it is striking that all central neurons simultaneously carry velocity, acceleration and jerk signals, with the relative contributions of each component depending on stimulus direction. Remarkably, we observed moderate further spatio-temporal processing (other than differences and response amplitude and baseline firing rate) beyond the VN/CN, such that roughly similar properties characterize all the cortical representations of vestibular translation signals examined to date.

Previous studies (*Rigotti et al., 2013*; *Fusi et al., 2016*) have proposed a role for the mixed non-linear responses widely observed in prefrontal cortex. It has been proposed that mixed selectivity plays an important computational role: high-dimensional representations with mixed selectivity allow a simple linear readout to generate a diverse array of potential responses (*Fusi et al., 2016*). In contrast, representations based on highly specialized neurons are low dimensional and may preclude a linear readout from generating several responses that depend on multiple task-relevant variables. Complex non-linear operations can be performed by simple linear summations of non-linear neurons, and in turn linear summation may be easily learned by neuronal networks. Thus, a rich variety of non-linearly transformed signals may facilitate the learning of complex computations.

It is possible that the described spatio-temporal transformations in VN/CN might reflect the need to perform non-linear, three-dimensional spatio-temporal operations necessary to implement an internal model of head motion (*Merfeld, 1995*; *Laurens and Angelaki, 2011*), whose influence has been documented in subcortical neurons (*Angelaki et al., 2004*; *Yakusheva et al., 2007*; *Laurens et al., 2013a*, *2013b*). Complex spatio-temporal representations of movement may be passed on to cortical (*Liu and Angelaki, 2009*; *Liu et al., 2011*) neurons to subserve their respective roles in spatial cognition. This hypothesis remains to be tested in future experiments.

## Materials and methods

### Subjects and apparatus

We include data from a total of 19 rhesus monkeys (Macaca mulatta). Each animal was chronically implanted with an eye coil, a head-restraint ring, and a plastic recording grid that contains an array of holes through which guide tubes were passed for extracellular electrophysiological recordings. All surgical and experimental procedures were approved by the Institutional Animal Care and Use Committee at Washington University and Baylor College of Medicine and were performed in accordance with institutional and NIH guidelines.

### Data sample

The data sample analyzed in the present study includes neurons recorded from multiple cortical areas (PIVC VIP, VPS, MSTd and FEF) using epoxy-coated tungsten microelectrodes (1–2 MΩ). Basic methodology and neural response properties of these cortical areas can be found in previous publications: PIVC: *Chen et al. (2010)*, VPS: *Chen et al. (2011a)*, VIP: *Chen et al. (2011b)*, MSTd: *Gu et al. (2006)*, *2010*; *Takahashi et al. (2007)* and FEF: *Gu et al. (2016)*. In order to compare with cortical neurons, we also recorded new data from 'vestibular-only' neurons without eye movement-related activity in the VN/CN (5–7 MΩ impedance electrodes) and otolith afferents (18–20 MΩ) using identical stimulation protocols (see below). To localize areas VN and CN, we first identified the abducens nuclei bilaterally based on their characteristic burst-tonic activity during horizontal eye movements (for details see *Meng et al., 2005*; *Liu et al., 2013*). The vestibular nerve was isolated beneath the auditory meatus as it entered the brain, as detailed in previous publications (*Haque et al., 2004*; *Yu et al., 2012*, *2015*). Note that the VN/CN sample in the current study also includes some of the same neurons included in *Liu et al. (2013)*. Samples of at least 40 neurons per region were collected; a sample size of 27 OA was considered sufficient due to the homogeneity of responses across OA.

### Experimental protocols and analysis

Monkeys were seated comfortably in a primate chair, which was secured to a six-degree-of-freedom motion platform (MOOG 6DOF2000E). We examined each cell's 3D spatio-temporal tuning by recording neural responses while the animal was translated along each of 26 directions spaced evenly within a sphere. This stimulus set includes all combinations of movement vectors having eight different azimuth angles (0, 45, 90, 135, 180, 225, 270 and 315°; forward, backward, leftward and rightward movements correspond to 90, 270, 0 and 180° respectively) and three different elevation angles, 0° (the stereotaxic horizontal plane) and ±45°, for a subtotal of 8×3 = 24 directions, as well as two additional movement vectors with elevation angles of −90° and 90° corresponding to upward and downward directions, respectively. Each movement trajectory consisted of a Gaussian velocity profile (0.2 s standard deviation) with corresponding biphasic acceleration and triphasic jerk profiles.

The total displacement was 13 cm and the peak acceleration was 0.1 g. The frequency content of this stimulus is illustrated in *Figure 1B*. All movements originated from the center of the movement range of the motion platform, and the platform returned to this starting position during the 2 s inter-trial interval. Stimuli were presented in random order within a single block of trials (at least five repetitions).

Data (neural activity and the translational acceleration stimulus) were collected either in complete darkness or during fixation of a central, head-fixed target in an otherwise dark room. Previous findings have established that otolith afferents, VN, CN and PIVC neurons are not sensitive to visual stimuli or eye movements (for details see *Bryan and Angelaki, 2009*; *Liu et al., 2013*; *Chen et al., 2010*). For MSTd, VPS, VIP and FEF neurons, which are sensitive to eye movements and/or visual motion, we have previously verified that vestibular translation responses during fixation are similar to those in complete darkness (details and comparison figures can be found in: *Gu et al., 2006*; *Takahashi et al., 2007* and *Chowdhury et al., 2009* for MSTd, *Chen et al., 2011a* for VPS, *Chen et al., 2011b* for VIP and *Gu et al., 2016* for FEF).

Quantitative data analyses were performed off-line using custom-written scripts in Matlab (Math-Works), available at the following adress: https://github.com/JeanLaurens/Spatiotemporal_Dynamics. Peristimulus time histograms (PSTHs) were constructed for each direction of translation using 25 ms time bins and were smoothed with a Gaussian kernel ($\sigma$ = 100 ms). We applied both temporal response modulation and space-time structure criteria to determine if a cell had responses strong enough to be included in the present analyses (see *Chen et al. (2011a)* for details). For the temporal response criterion, we first found the peak (trough) time of the PSTH for each stimulus direction. Using the peak (trough) time, we obtained a distribution of response values from the PSTHs for each trial at that point in time. We then compared this peak (trough) distribution with the distribution of values obtained from baseline firing using the Wilcoxon signed-rank test. If responses to two neighboring stimulus directions (45° apart) had significantly ($p<0.01$) different peak (trough) distributions compared to their own baseline distributions, then the cell passed the temporal response modulation criterion. For the space-time structure criterion, we performed a two-way ANOVA on the full set of single-trial PSTHs, with space and time as factors (*Chen et al., 2011c*). Cells were considered to have space-time structure when both factors as well as their interaction were highly significant ($p<0.001$).

To ensure robustness, non-parametric statistics were used and median results were reported whenever it was possible. Confidence intervals of medians were computed based on bootstrap resampling.

## Spatiotemporal curve fitting

To characterize and quantify the spatiotemporal dynamics of vestibular responses, trial-averaged PSTHs for each direction of translation were fit with spatiotemporal functions of varying complexity. In its simplest single-component form, the fitted function consists of a spatial tuning component, $y(g(\theta,\phi))$ and a temporal response profile $f(t-\tau_0)$, multiplied together to obtain the spatiotemporal response function (*Figure 1*):

$$FR(\theta,\phi,t) = W_0 \cdot y(g(\theta,\phi)) \cdot f(t-\tau_0) + FR_0 \tag{1}$$

where $W_0$ is the response amplitude and $FR_0$ is the baseline firing rate. Note that $y(g(\theta,\phi))$ and $f(t-\tau_0)$ are normalized and unitless, such that $W_0$ and $FR_0$ are both expressed in spikes/s. Other parameters are defined below.

***The spatial tuning*** is modelled as a cosine function $g(\theta,\phi)$ fed through a non-linear function $y(x)$. The cosine tuning function $g(\theta,\phi)$ depends on parameters $\theta_0$ (preferred azimuth) and $\phi_0$ (preferred elevation):

$$g(\theta,\phi) = r(\theta,\phi)r(\theta_0,\phi_0)^\top \tag{2}$$

where $r(\theta,\phi) = [cos(\theta).cos(\phi), sin(\theta).cos(\phi), sin(\phi)]$ maps the unit vector in spherical coordinates to Cartesian coordinates (*Mardia and Jupp, 1999*). Because $g(\theta,\phi)$ is equal to the cosine of the angle between $r(\theta,\phi)$ and $r(\theta_0,\phi_0)$, it has a value of 1 at the preferred direction (PD), a value of $-1$ at the opposite direction (called anti-preferred direction) and a value of 0 in any direction orthogonal to the PD (called 'an orthogonal direction').

The non-linear function, $y(x)$ generalizes the spatial tuning from a pure cosine function by adding an offset parameter, $o_0$. It takes the form

$$y(x) = o_0 + (1 - |o_0|).x \qquad (3)$$

The offset parameter, $o_0$, ranging from $-1$ to $1$, allows us to model omnidirectional response components, while the term $(1 - |o_0|).x$ represents cosine tuning as a function of motion direction. Note that scaling $x$ by $(1 - |o_0|)$ ensures that the function $y(x)$ is normalized, such that $|y(x)| \leq 1$ for all $x$ in $[-1\ 1]$ and $|y(x)| = 1$ for at least one value of $x$.

**The temporal profile**, $f(t)$, is defined as follows:

$$\text{Velocity}: \qquad f_v(t - \tau_0) = \left\lceil exp\left(\frac{-(t - \tau_0)^2}{2\sigma^2}\right)\right\rceil^1 \qquad (4)$$

$$\text{Acceleration}: \qquad f_a(t - \tau_0) = \left\lceil \frac{\tau_0 - t}{\sigma^2} exp\left(\frac{-(t - \tau_0)^2}{2\sigma^2}\right)\right\rceil^1 \qquad (5)$$

$$\text{Jerk}: \qquad f_j(t - \tau_0) = \left\lceil \frac{(t - \tau_0)^2 - \sigma^2}{\sigma^4} exp\left(\frac{-(t - \tau_0)^2}{2\sigma^2}\right)\right\rceil^1 \qquad (6)$$

In **Equations 4–6**, $\tau_0$ (temporal delay) is a fitted parameter, whereas $\sigma$ (temporal Gaussian width) is set by the stimulus ($\sigma$ = 0.2 s). The operator $\lceil . \rceil^1$ indicates that the temporal profiles were normalized so that the difference between the maximum and the minimum values of each profile is 1 (**Figure 1**, blue, red and green waveforms, respectively).

The simplest spatio-temporal model consisted of Velocity-only ('V' model), Acceleration-only ('A' model) or Jerk-only ('J' model) terms based on **Equation 1**, where the temporal profile was given by **Equations 4, 5 or 6**, respectively. The most general model was the Velocity+Acceleration+Jerk (VAJ) model, given by the following equation:

$$FR(\theta, \phi, t) = W_v y_v(g_v(\theta, \phi))f_v(t - \tau_0) + W_a y_a(g_a(\theta, \phi))f_a(t - \tau_0) + W_j y_j(g_j(\theta, \phi))f_j(t - \tau_0) + FR_0 \qquad (7)$$

where $y_v(g_v(\theta, \phi))f_v(t - \tau_0)$, $y_a(g_a(\theta, \phi))f_a(t - \tau_0)$ and $y_j(g_j(\theta, \phi))f_j(t - \tau_0)$ are the velocity, acceleration and jerk components, respectively. Each of the three spatial tuning functions $(y_v(g_v(\theta, \phi)), y_a(g_a(\theta, \phi)), y_j(g_j(\theta, \phi)))$ has its own set of parameters $(\theta_v, \varphi_v, o_v, \theta_a, \varphi_a, o_a$ and $\theta_j, \varphi_j, o_j$, respectively), and $W_v$, $W_a$ and $W_j$ are the respective weights on each component. Note that a single delay parameter $\tau_0$ is used in the VAJ model. Additional models of intermediate complexity—Velocity + Acceleration ('VA'), Velocity + Jerk ('VJ'), Acceleration + Jerk ('AJ')—were also tested. The 1-component models ('V', 'A' or 'J') have six free parameters, the two-component models ('VA', 'VJ' or 'AJ') have 10 free parameters, and the three-component model ('VAJ') has 14 parameters.

How well each model fit the neural responses was quantified as the proportion of variance accounted for by the model ($R^2$), and was computed by regressing the responses of each neuron against the values of each fitted function (across the 26 heading directions and the entire 2 s response profile). To evaluate the best model while accounting for the number of model parameters, we used the Bayesian Information Criterion (BIC), defined as:

$$BIC = n\log\frac{RSS}{n} + p\log(n) \qquad (8)$$

where $RSS$ is the residual sum of squares, $n$ is the number of data points (considering that the response profile was 2 s long and filtered by a Gaussian kernel with $\sigma$ = 100 ms, we assumed that the data amounted to 10 independent time points per profile and therefore n = 260) and $p$ is the number of function parameters (**Konishi and Kitagawa, 2008**). The best model based on this criterion is the one with the lowest BIC value.

The importance of each model component was also assessed by computing its partial coefficient of correlation given the two other components. For instance, the partial correlation coefficient of the A component $R_{A|VJ}$ reflects how much variance is explained by adding the A component to the VJ model and is computed accorded to:

$$R^2_{A|VJ} = \left(R^2_{VAJ} - R^2_{VJ}\right)/\left(1 - R^2_{VJ}\right) \tag{9}$$

The VAJ model includes a separate spatial tuning curve for each dynamic component (acceleration, velocity and jerk). However, for certain cells, the dynamic components have an identical spatial tuning; a property called 'spatio-temporal separability'. We tested if the tuning of cells was separable by fitting an additional 'Separable VAJ' model (which has eight parameters) that includes an identical spatial tuning curve for all components. This model was used to compute a 'separability index, (Sep I)':

$$\mathrm{Sep\,I} = R^2_{\mathrm{Separable\,VAJ}}/R^2_{\mathrm{VAJ}} \tag{10}$$

In order to validate our approach, we simulated 1000 Poisson-spiking neurons, divided equally among each model type, and verified that fitting the standard model could satisfactorily retrieve the spatio-temporal tuning of neurons.

Finally, we also tested a four-component (PVAJ) model (18 parameters), in which a position (integral of velocity) component was added to the other three terms (velocity, acceleration and jerk). However, the recorded data used to evaluate these models were limited to at most 200 ms following motion offset, which makes the contribution of position modulation hard to evaluate reliably. For this reason, we focus mainly on the findings of the VAJ model (but see *Figure 10*).

## Acknowledgements

The work was supported by NIH grants EY12814 and DC04260. GCD was supported by EY016178. The authors would like to thank A Chen, Y Gu, A Bryan for recording the central neurons whose properties have been analyzed here.

## Additional information

### Funding

| Funder | Grant reference number | Author |
|---|---|---|
| National Eye Institute | EY12814 | Jean Laurens<br>Sheng Liu<br>Xiong-Jie Yu<br>David Dickman<br>Dora E Angelaki |
| National Institute on Deafness and Other Communication Disorders | DC04260 | Jean Laurens<br>Sheng Liu<br>Xiong-Jie Yu<br>David Dickman<br>Dora E Angelaki |
| National Eye Institute | EY016178 | Gregory C DeAngelis |

The funders had no role in study design, data collection and interpretation, or the decision to submit the work for publication.

### Author contributions

JL, Formal analysis, Writing—original draft, Writing—review and editing; SL, Investigation, Writing—original draft, Writing—review and editing; X-JY, Investigation, Writing—original draft; RC, Data curation, Formal analysis, Writing—original draft; DD, Conceptualization, Supervision, Writing—original draft, Writing—review and editing; GCD, Conceptualization, Supervision, Methodology, Writing—original draft, Writing—review and editing; DEA, Conceptualization, Supervision, Funding acquisition, Investigation, Writing—original draft, Writing—review and editing

### Author ORCIDs

Jean Laurens, http://orcid.org/0000-0002-9101-2802
Gregory C DeAngelis, http://orcid.org/0000-0002-1635-1273
Dora E Angelaki, http://orcid.org/0000-0002-9650-8962

## Ethics

Animal experimentation: This study was performed in strict accordance with the recommendations in the Guide for the Care and Use of Laboratory Animals of the National Institutes of Health. All of the animals were handled according to approved institutional animal care and use committee (IACUC) protocol #AN-5795 of the Baylor College of Medicine. Data collected in previous studies were obtained according to the National Institutes of Health Guidelines, as described in the corresponding publications. All surgery was performed under isoflurane anesthesia, and every effort was made to minimize suffering.

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
