## [Decision Letter]

Thank you for submitting your article "Evolution of spatiotemporal dynamics from otolith sensory afferents to cortex" for consideration by *eLife*. Your article has been reviewed by three peer reviewers, one of whom, Jennifer L Raymond (Reviewer #1), is a member of our Board of Reviewing Editors and the evaluation has been overseen by David Van Essen as the Senior Editor.

The reviewers have discussed the reviews with one another and the Reviewing Editor has drafted this decision to help you prepare a revised submission.

Summary:

This paper characterizes the spatial and temporal coding properties of neurons throughout the vestibular system hierarchy. This work may be unprecedented in its comprehensive and systematic characterization of neurons at so many different levels of a sensory system, using a common set of stimuli and analysis methods. 579 neurons were recorded in 8 subcortical and cortical regions carrying vestibular signals (otolith afferents, vestibular and cerebellar nuclei, and 5 cortical areas: PIVC, VIP, VPS, MSTd, FEF) during transient, translational vestibular stimuli delivered in 24 directions in 3D space. Neural responses were fit with a model that included velocity, acceleration and jerk, and linear or nonlinear spatio-temporal combinations of these quantities. The analysis is principled, systematic, uniformly applied across brain areas, and validated using simulated neurons with defined coding properties. The authors find that whereas there are clear differences in response selectivity between of primary otolith afferents and central neurons, responses of neurons in vestibular and cerebellar nuclei and five distinct cerebrocortical areas are remarkably similar to each other. The finding of similar spatio-temporal coding properties in all brain regions beyond the otolithic afferents suggests that a key step in the processing of vestibular information is performed at the first synapse. Overall, the work is sound and carefully executed. It is impressive in its completeness, providing a comprehensive picture of vestibular signal transformations across brain regions. The central findings provide a valuable resource for the vestibular field, and provide the opportunity for useful comparisons across sensory modalities. Although generally enthusiastic, the Reviewers had several suggestions for additional analysis, and for how the presentation of the Results and Discussion could be improved to increase the impact of this impressive body of research.

Essential revisions:

1) The main conclusion, that the big signal transformation happened at the first processing stage and that all the higher vestibular areas are more similar in their coding properties, is well supported. However, the difficulty for the reader to draw any conclusion beyond this creates an impression that not much was gained from the massive effort invested. A more nuanced treatment of differences between central brain areas seems merited, and would do the rich data set more justice. Some specific examples, raised by the Reviewers:

a) A previous study from this group (Yakusheva et al., J Neurosci 2013) had shown that vestibular responses of Purkinje cells in the nodulus/ ventral uvula were similar to those of MSTd, but not PIVC or VIP neurons. Purkinje cells are not recorded here, but it would be useful to cite this previous study to provide context for the current results, especially since the take-home message appears to be quite different (MSTd and Purkinje neurons were similar to each other but different from PIVC and VIP neurons in Yakusheva et al. 2013, but in the current manuscript the argument is that all central areas are more or less equivalent).

b) The ternary plots in Figure 6 seem to emphasize similarities between regions. For example, VIP and MSTd are plotted together in C, and this and the superposition of the envelopes of all of the data in D obscures the fact that these two regions actually have quite distinct velocity weights. Similarly, there are many more neurons in 6C (representing MSTd, VIP, FEF) that have 0 acceleration weights (fall along the bottom of the triangle) than in the other plots/ regions, but the envelope representation in 6D obscures that fact. Comparing this (Figure 6) approach to data visualization vs. the approach in Figure 10, it is not clear to me whether different conclusions about variability across regions are not reached partly because of differences in data analysis/ presentation. In Figure 10, where actual histograms are presented for position components for each region, the conclusion reached is quite different – that only MSTd has a sizable position component. Would we more easily pick out differences across regions in the velocity/ acceleration/ jerk weights of Figure 6 if they had been represented this way (or with cumulative histograms for each of the 3 weights, as in Figure 9—figure supplement 1)?

c) The text describing Figure 7 says that central brain areas have lower separability indices. Although this is true for OAs vs higher areas overall, in moving from the vestibular nuceli (VN) more centrally, there is a progressive increase in the separability index. The central cells also have more variability in their separability index than the OAs. Some comment about this seems warranted--the existing text provides too superficial consideration of the results.

2) The general message is that "there is a bit of everything everywhere" with only marginal differences across regions; but the complicated analysis (many parameters reported one by one, or 3 at a time Figure 6) and incomplete/indirect reporting (see point 3 below) leaves open another possibility: might there be clusters in the ~ 17 dimension parameter space, which would show a stronger segregation across brain regions?

3) In general, the analysis was well done and well presented, but it might be improved in several ways.

a) This model is "instantaneous", meaning that the rate is related to an instantaneous value of the movement (velocity, acceleration, jerk), and that the history of cell firing is not influencing the firing rate. The three sets of curves (gaussian profile and the 2 first derivative) provide a good set of functions of time for the decomposition of PSTHs: e.g. for a simple profile with a peak, the mean/sd of the peak will correspond to the speed, the assymetry (skewness) will correspond to the acceleration, and the sharpness of the peak (kurtosis) will correspond to the jerk. The shape of the PSTH may indeed be conditioned by the sensitivity of a cell to the speed/acceleration/jerk, but alternate causes, such as long-term impact of movement parameters or of past cellular activity (i.e. violation from the "instantaneous hypothesis"), might also explain the shape of the PSTH; indeed, this is observed in many examples of the Figure 3, where significant departure from baseline are observed at the end of the psth, when virtually all movement is over for hundreds of milliseconds: see Figure 3 (e.g. end of the PSTH Az 315 deg, Elev 45 deg) or Figure 3—figure supplement 1 (e.g. end of Az 270, Elev -45), Figure 3—figure supplement 2 (e.g. Az 90, Elev 0 deg), Figure 3—figure supplement 3 (end of Az 315, Elev 45). All these example suggest that to some degree, the firing of the cells may be affected by past events (movement or cell activity) at much longer time scales than the few tens to hundred milliseconds used for the delay. Therefore, the correspondence between the weights of the fits to a gaussian profile and its first derivatives and an actual sensitivity to velocity, acceleration and jerk is somewhat questionable.

b) A second concern is linked to the use of a non-linear transformation: to describe the spatial dependency of the PSTHs, the sensitivity to the movement components (velocity, acceleration, jerk) is tuned by a non-linear transformation of the cosine of the angle between a (fitted) preferred direction -which may vary across components- and the actual movement direction. The description of these non-linear function is remarkably obfuscated (described by three successive set of variables: a,b,c in y(x)=a.b^x+c, then by y(-1),y(0),y(1), then by Z0 and k0….), but the end result is more clearly shown in Figure 9. One of the important points is that these non-linear functions are not mapping the cosine result (range [-1,1]) onto the full [-1,1] range, but onto a subset of this range. Then, the fit predicts the (spatial dependency of the) rate based on a weighted sum of the non-linearly transformed cosine function. In the MS, theses weights of the components of the fit are used to compare the sensitivities of the cells (e.g. Figure 6). This procedures calls for a few comments:

First, it is unclear if non-linearity is really necessary: inspection of the classes of the 'non-linear functions' (Figure 9) shows set of profiles close to linearity which could be as well approximated by linear (ax+b) functions; the main difference across classes seems more on the target interval than on the shape. The non-linear version will always do better since it permits a finer tuning (1 more parameter), but the benefit may be only marginal (I think the authors could reasonably easily test this) and come at the price a quite obscure process.

Second, because the range over which the cosine is mapped by the non-linear function may be much smaller than the -1,1 range, the weights are not reflecting the two qualitative aspects of the spatial tuning: for example, if a cell has a large gaussian PSTH in all direction, with a 10% modulation of the amplitude of the peak, this will show up as a strong weight of velocity (and a 'broad-positive' non linear function), while it would indeed better described as a strong sensitivity to speed (length of the velocity vector) but a weak sensitivity to velocity (as a vector). In this case, the large weight of the velocity would correspond only to a small dynamic range over spatial orientations; the same large weight for another cell with cosine tuning would correspond to a full dynamic range over spatial orientations. Therefore, a single value of the weight is mixing qualitatively different information (e.g. speed value as in Figure 3—figure supplement 3, versus velocity vector as in Figure 3—figure supplement 4). The only things that disambiguate the information are the parameters of the non-linear function, but 1) they are not so trivial (the authors found it is easier to report them as 'arbitrary' categories…), 2) they are reported independently from the values of the weight. A linear version of the transformation ('a.cos(x)+b', provided that it would not degrade too much the performance of the fits) could be easier to report and avoid the categorization: 'a' (in the a.cos(x)+b) would signal the dynamic range over 3D, and 'b' would signal the directionaly-invariant component of the response; it would become possible for example to generate bivariate plots (such as 'weight' vs 'a') which would answer simple questions such as: is the dynamic range for velocity coding identical in all brain areas for strongly velocity-tuned cells? Currently, the choice of reporting makes the answer to this question unclear. Since some of their conclusion is that there are no regional differences, I do not feel so certain they took the most powerful approach to reveal them.

c) It is not clear why the color-coded stacked bar plots are used in Figure 7, and Figure 10 – here the color map is representing PD-angles, which presumably vary continuously, but they have been categorized unnecessarily – cumulative histograms, or at least a continuous color map, here could allow more direct comparisons across regions.

4) The discussion could be more enlightening:

General interest could be broadened by a brief comparison with other sensory systems

Why would the brain maintain such complicated representation of movement (a parameter and its two first derivatives or its integral, with unrelated preferred directions in many cases and non-linear spatial sensitivity to departure from preferred direction) across all brain areas, without further processing beyond the first synapse?

The last paragraph is obscure: what would be the relation between tilt/translation disambiguation and the mixtures of sensitivity described in the study?

Some of the big picture, clearly laid out in the Introduction, is lost in the Results and Discussion sections. For example, it would be useful for the authors to remind us of why we care about space time separability

---

## [Author Response]

*Essential revisions:*

*1) The main conclusion, that the big signal transformation happened at the first processing stage and that all the higher vestibular areas are more similar in their coding properties, is well supported. However, the difficulty for the reader to draw any conclusion beyond this creates an impression that not much was gained from the massive effort invested. A more nuanced treatment of differences between central brain areas seems merited, and would do the rich data set more justice.*

It was indeed a massive effort to do this analysis that took many years and multiple people. As summarized below, following the reviewers’ comments, we now re-ran the whole analysis again, and the results have been simplified significantly (thank you reviewers!). We have also tried to better highlight the main conclusions: (1) large differences between brainstem and otolith afferents, (2) similar spatio-temporal properties in central areas – however, there are systematic differences in response amplitude and spontaneous firing rate (3) strong presence of both velocity and jerk tuning in central neurons, (4) strongly non-linear tuning, particularly for velocity and jerk in central neurons, (5) position-tuning mostly in MSTd. We have also substantially beefed up the Discussion to address the reviewers’ concerns.

*Some specific examples, raised by the Reviewers:*

*a) A previous study from this group (Yakusheva et al., J Neurosci 2013) had shown that vestibular responses of Purkinje cells in the nodulus/ ventral uvula were similar to those of MSTd, but not PIVC or VIP neurons. Purkinje cells are not recorded here, but it would be useful to cite this previous study to provide context for the current results, especially since the take-home message appears to be quite different (MSTd and Purkinje neurons were similar to each other but different from PIVC and VIP neurons in Yakusheva et al. 2013, but in the current manuscript the argument is that all central areas are more or less equivalent).*

We have now added this reference, and also pointed out the differences between central regions. Most importantly, the present analysis is not directly comparable with the previous analysis, which was restricted to motion in a single plane, did not include the jerk component, and used a different form of nonlinearity. Yet, results in this paper and in the previous analyses (including Chen et al. 2011c) are rather consistent. This is now better highlighted in the Discussion.

*b) The ternary plots in Figure 6 seem to emphasize similarities between regions. For example, VIP and MSTd are plotted together in C, and this and the superposition of the envelopes of all of the data in D obscures the fact that these two regions actually have quite distinct velocity weights. Similarly, there are many more neurons in 6C (representing MSTd, VIP, FEF) that have 0 acceleration weights (fall along the bottom of the triangle) than in the other plots/ regions, but the envelope representation in 6D obscures that fact. Comparing this (Figure 6) approach to data visualization vs. the approach in Figure 10, it is not clear to me whether different conclusions about variability across regions are not reached partly because of differences in data analysis/ presentation. In Figure 10, where actual histograms are presented for position components for each region, the conclusion reached is quite different – that only MSTd has a sizable position component. Would we more easily pick out differences across regions in the velocity/ acceleration/ jerk weights of Figure 6 if they had been represented this way (or with cumulative histograms for each of the 3 weights, as in Figure 9—figure supplement 1)?*

We thank the reviewer for this suggestion. We found that cumulative distribution plots did indeed provide a convenient way to compare brain areas, so we have added cumulative plots to Figure 6. We have also modified the ternary plots, as described next.

*c) The text describing Figure 7 says that central brain areas have lower separability indices. Although this is true for OAs vs higher areas overall, in moving from the vestibular nuceli (VN) more centrally, there is a progressive increase in the separability index. The central cells also have more variability in their separability index than the OAs. Some comment about this seems warranted--the existing text provides too superficial consideration of the results.*

Indeed, we found that VN and CN have lower separability indexes than other brain regions. We have found that the PD of the 3 components are less aligned in VN and CN compared to other brain regions, which explains the lower separability, and we now describe this in the text.

*2) The general message is that "there is a bit of everything everywhere" with only marginal differences across regions; but the complicated analysis (many parameters reported one by one, or 3 at a time Figure 6) and incomplete/indirect reporting (see point 3 below) leaves open another possibility: might there be clusters in the ~ 17 dimension parameter space, which would show a stronger segregation across brain regions?*

We have conducted a variety of cluster analyses as well as factor analyses. We found that the results of these analyses resemble the results described here. The OAs stand out from other cells very clearly, while the other neuronal populations tend to overlap in a single cloud of dots. MSTd, which tends to have the highest velocity responses, tends to be somewhat offset from, but still overlaps largely with the other regions. Thus, as this cluster analysis does not add anything new, we have not included it in the manuscript. We hope that we managed to simplify the reporting of our results, in particular by adopting the simpler non-linearity suggested by the reviewers (where the model with the largest number of parameters now has 14, not 17).

*3) In general, the analysis was well done and well presented, but it might be improved in several ways.*

*a) This model is "instantaneous", meaning that the rate is related to an instantaneous value of the movement (velocity, acceleration, jerk), and that the history of cell firing is not influencing the firing rate. The three sets of curves (gaussian profile and the 2 first derivative) provide a good set of functions of time for the decomposition of PSTHs: e.g. for a simple profile with a peak, the mean/sd of the peak will correspond to the speed, the assymetry (skewness) will correspond to the acceleration, and the sharpness of the peak (kurtosis) will correspond to the jerk. The shape of the PSTH may indeed be conditioned by the sensitivity of a cell to the speed/acceleration/jerk, but alternate causes, such as long-term impact of movement parameters or of past cellular activity (i.e. violation from the "instantaneous hypothesis"), might also explain the shape of the PSTH; indeed, this is observed in many examples of the Figure 3, where significant departure from baseline are observed at the end of the psth, when virtually all movement is over for hundreds of milliseconds: see Figure 3 (e.g. end of the PSTH Az 315 deg, Elev 45 deg) or Figure 3—figure supplement 1 (e.g. end of Az 270, Elev -45), Figure 3—figure supplement 2 (e.g. Az 90, Elev 0 deg), Figure 3—figure supplement 3 (end of Az 315, Elev 45). All these example suggest that to some degree, the firing of the cells may be affected by past events (movement or cell activity) at much longer time scales than the few tens to hundred milliseconds used for the delay. Therefore, the correspondence between the weights of the fits to a gaussian profile and its first derivatives and an actual sensitivity to velocity, acceleration and jerk is somewhat questionable.*

From a purely descriptive point of view, responses that persist through the end of the trial may be explained by sensitivity to position that has been modeled by the PVAJ model (Figure 10). However, we also acknowledge that our model is essentially descriptive and not mechanistic. Therefore, the dynamic responses that we attribute to velocity or jerk could be the result of non-linear dynamic processes of a more complex nature than simple integration or differentiation. In other words, our model is limited to describing spatiotemporal dynamics as a combination of basis functions that are related to the stimulus, rather than as a combination of inputs to each neuron. While the latter type of model may be possible, we feel that it would be a major undertaking and may not be sufficiently constrained to be successful. Despite this limitation (which is now added to the Discussion), we think that our approach still adequately characterizes the nature of the transformations of vestibular signals.

*b) A second concern is linked to the use of a non-linear transformation: to describe the spatial dependency of the PSTHs, the sensitivity to the movement components (velocity, acceleration, jerk) is tuned by a non-linear transformation of the cosine of the angle between a (fitted) preferred direction -which may vary across components- and the actual movement direction. The description of these non-linear function is remarkably obfuscated (described by three successive set of variables: a,b,c in y(x)=a.b^x+c, then by y(-1),y(0),y(1), then by Z0 and k0….), but the end result is more clearly shown in Figure 9.*

We have completely changed (and thus greatly simplified) the manuscript based on these suggestions. Thank you! (a great example of a useful review process!)

*One of the important points is that these non-linear functions are not mapping the cosine result (range [-1,1]) onto the full [-1,1] range, but onto a subset of this range. Then, the fit predicts the (spatial dependency of the) rate based on a weighted sum of the non-linearly transformed cosine function. In the MS, theses weights of the components of the fit are used to compare the sensitivities of the cells (e.g. Figure 6). This procedures calls for a few comments:*

*First,, it is unclear if non-linearity is really necessary: inspection of the classes of the 'non-linear functions' (Figure 9) shows set of profiles close to linearity which could be as well approximated by linear (ax+b) functions; the main difference across classes seems more on the target interval than on the shape. The non-linear version will always do better since it permits a finer tuning (1 more parameter), but the benefit may be only marginal (I think the authors could reasonably easily test this) and come at the price a quite obscure process.*

We thank the reviewer for this suggestion. We fitted an affine function, and found, indeed, very similar results (as described in Figure 4—figure supplement 1). We are now using these new fits throughout the manuscript. Note that (ax+b) is not a linear function but an affine function (in linear algebra, two necessary conditions for a function f(x) to be linear are f(0)=0 and f(a+b)=f(a)+f(b)) and therefore this function is still referred to as non-linear through the text.

*Second, because the range over which the cosine is mapped by the non-linear function may be much smaller than the -1,1 range, the weights are not reflecting the two qualitative aspects of the spatial tuning: for example, if a cell has a large gaussian PSTH in all direction, with a 10% modulation of the amplitude of the peak, this will show up as a strong weight of velocity (and a 'broad-positive' non linear function), while it would indeed better described as a strong sensitivity to speed (length of the velocity vector) but a weak sensitivity to velocity (as a vector). In this case, the large weight of the velocity would correspond only to a small dynamic range over spatial orientations; the same large weight for another cell with cosine tuning would correspond to a full dynamic range over spatial orientations. Therefore, a single value of the weight is mixing qualitatively different information (e.g. speed value as in Figure 3—figure supplement 3, versus velocity vector as in Figure 3—figure supplement 4). The only things that disambiguate the information are the parameters of the non-linear function, but 1) they are not so trivial (the authors found it is easier to report them as 'arbitrary' categories…), 2) they are reported independently from the values of the weight. A linear version of the transformation ('a.cos(x)+b', provided that it would not degrade too much the performance of the fits) could be easier to report and avoid the categorization: 'a' (in the a.cos(x)+b) would signal the dynamic range over 3D, and 'b' would signal the directionaly-invariant component of the response; it would become possible for example to generate bivariate plots (such as 'weight' vs 'a') which would answer simple questions such as: is the dynamic range for velocity coding identical in all brain areas for strongly velocity-tuned cells? Currently, the choice of reporting makes the answer to this question unclear. Since some of their conclusion is that there are no regional differences, I do not feel so certain they took the most powerful approach to reveal them.*

We have followed the reviewer’s suggestions, for which we are thankful. More precisely, we now parametrize the non-linear function as y(x) = o + (1-|o|).cos(x) (ensuring that the function is normalized, such tht |y(x)|<=1 and y(-1)=-1 or y(1)=1). Next we took the weight w into account and plotted w.o as a function of w.(1-|o|). The term w.o represents the directionally-invariant component (we called it ‘offset’) and the term w.(1-|o|) represents the dynamic range (we called it ‘cosine tuning’).

*c) It is not clear why the color-coded stacked bar plots are used in Figure 7, and Figure 10 – here the color map is representing PD-angles, which presumably vary continuously, but they have been categorized unnecessarily – cumulative histograms, or at least a continuous color map, here could allow more direct comparisons across regions.*

We have introduced continuous color maps in these plots and also reversed the green and red colors (red colors are now indicating aligned PD).

*4) The Discussion could be more enlightening:*

*General interest could be broadened by a brief comparison with other sensory systems*

*Why would the brain maintain such complicated representation of movement (a parameter and its two first derivatives or its integral, with unrelated preferred directions in many cases and non-linear spatial sensitivity to departure from preferred direction) across all brain areas, without further processing beyond the first synapse?*

*The last paragraph is obscure: what would be the relation between tilt/translation disambiguation and the mixtures of sensitivity described in the study?*

*Some of the big picture, clearly laid out in the Introduction, is lost in the Results and Discussion sections. For example, it would be useful for the authors to remind us of why we care about space time separability*

We have tried to re-work and improve the Discussion. We rewrote the last paragraph to provide a clearer hypothesis about the role of non-linear encoding of motion, and referred to previous work of ‘mixed selectivity’ in the prefrontal and parietal cortex.